# Towards Reasonable Concept Bottleneck Models

## Abstract

We propose a novel, flexible, and efficient framework for designing Concept Bottleneck Models (CBMs) that enables practitioners to explicitly encode and extend their prior knowledge and beliefs about the concept-concept (C-C) and concept-task (C→Y) relationships within the model's reasoning when making predictions. The resulting **C**oncept **REA**soning **M**odels (CREAMs) architecturally encode arbitrary types of C-C relationships such as mutual exclusivity, hierarchical associations, and/or correlations, as well as potentially sparse C→Y relationships. Moreover, CREAM can optionally incorporate a regularized side-channel to complement the potentially incomplete concept sets, achieving competitive task performance while encouraging predictions to be concept-grounded. To evaluate CBMs in such settings, we introduce a C→Y agnostic metric that quantifies interpretability when predictions partially rely on the side-channel. In our experiments, we show that, without additional computational overhead, CREAM models support efficient interventions, can avoid concept leakage, and achieve black-box-level performance under missing concepts. We further analyze how an optional side-channel affects interpretability and intervenability. Importantly, the side-channel enables CBMs to remain effective even in scenarios where only a limited number of concepts are available.

## 1 Introduction

Deep neural networks (DNNs) have become ubiquitous in various aspects of our daily lives but their opaque decision-making limits transparency, user understanding, and trust. Interpretability is essential for reliable AI, especially in finance (Doshi-Velez & Kim, 2017), healthcare (Rudin, 2019), autonomous systems (Doshi-Velez & Kim, 2017; Lipton, 2018; Samek et al., 2019), and so forth.

Interpretable models have therefore gained attention (Molnar, 2025; Rudin et al., 2022), particularly concept-based approaches that explain predictions through human-understandable concepts (Barbiero et al., 2023; Chen et al., 2020; Koh et al., 2020; Oikarinen et al., 2023; Poeta et al., 2023; Yeh et al., 2020; Yuksekgonul et al., 2023). Concept Bottleneck Models (CBMs) (Koh et al., 2020) exemplify this by introducing an intermediate concept layer, where concepts are explicitly learned and predicted prior to the task, enabling *transparent reasoning* and *human intervention*. CBMs have been applied on various fields, including medical diagnosis (Daneshjou et al., 2022), predictive maintenance (Forest et al., 2024), and vision-language tasks (Yang et al., 2023).

Standard CBMs assume conditional independence among concepts, limiting their ability to model intra-concept or concept–task relationships (Dominici et al., 2025), a property we call *structured model reasoning*. They also assume the concept set is complete and sufficient for prediction. Extensions have relaxed these assumptions (Dominici et al., 2025), they typically do so in problem-specific ways, introducing new constraints such as restrictions to particular equation families or reasoning structures (e.g., directed acyclic graphs). Real-world datasets often exhibit *concept incompleteness*, which reduces accuracy (Grivas et al., 2024; Yeh et al., 2020). Moreover, even with correct concept predictions, models may exploit unintended information, called *concept leakage*, to bypass intended reasoning pathways (Mahinpei et al., 2021; Margeloiu et al., 2021), undermining interpretability and encouraging misplaced trust (Marconato et al., 2023a).

We propose **C**oncept **REA**soning **M**odels (CREAM), a framework for CBMs that encodes prior knowledge of C-C and C→Y relations through a reasoning graph. The graph combines hard constraints (e.g., blocking

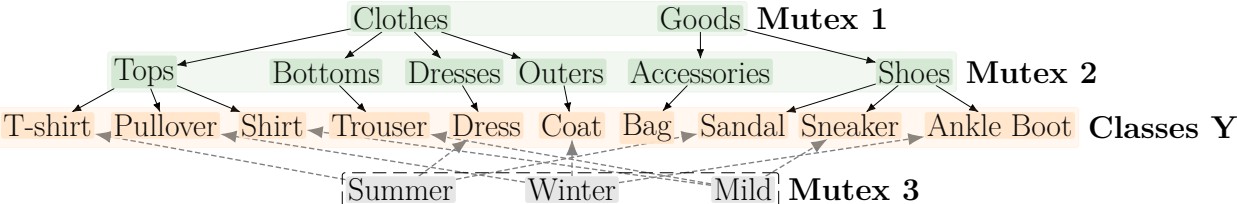

Figure 1: Reasoning graph for FMNIST from (Seo & Shin, 2019). We show the incomplete concept set used in *iFMNIST* as well as the additional season-related concepts from *cFMNIST*. Concepts and classes within the boxes are mutually exclusive. Edges between concepts represent C-C relationships, while edges from concepts to classes represent C→Y dependencies.

irrelevant edges, enforcing exclusivity) with probabilistic dependencies learned from data. By embedding these relations into CREAM, predictions are grounded in a user-specified reasoning graph. Each concept influences only a sparse subset of predictions and any given output can be traced back to a limited set of candidate concepts, ensuring tractability and enhancing interpretability and intervenability while mitigating leakage. A regularized side-channel captures supplementary task-relevant information without compromising concept-based predictions, unlike prior work (Dominici et al., 2025). Importantly, the C-C, C→Y blocks, and side-channel are independent modular components that can each be modified, included, or excluded separately, depending on the available knowledge and assumptions. CREAM accommodates alternative approaches under different assumptions about the reasoning graph, supporting modular designs aligned with available knowledge. Overall, it provides an interpretable, modular and adaptable framework balancing predictive accuracy with controlled reasoning.

## 2 Related Work

**Connection to Neurosymbolic approaches.** CBMs are complementary to Neurosymbolic (NeSy) approaches (Badreddine et al., 2022; Bortolotti et al., 2024; Manhaeve et al., 2018; Marconato et al., 2023a); the former require supervision solely on the concepts while the latter require knowledge usually in the form of logical programs. In our case, we require knowledge about *(directed) statistical (in)dependencies* between interpretable variables, to constrain the relationships between them. In App. H we provide a logic-based viewpoint to CREAM. For instance, a mutual exclusivity constraint would be described as: Clothes $\sqcap$ Goods $\sqsubseteq \perp$, and concepts would be calculated by: Tops $\leftarrow \mathbf{z}_{Clothes} \sqcap \mathbf{z}_{Tops}$.

**Concept and Task Relationships.** Standard CBMs assume that concepts are *independent* and that all of them directly contribute to the task, thus forming a bipartite graph ($G$). To address this, several works have incorporated concept interdependencies. Relational CBMs (Barbiero et al., 2024) use graph-structured data and message-passing algorithms to propagate relational dependencies, while Stochastic CBMs (SCBMs) (Vandenhirtz et al., 2024) model concepts using a learnable covariance matrix. Similarly, Autoregressive CBMs (ACBMs) (Havasi et al., 2022) introduce an autoregressive structure to learn sequential dependencies between concepts. These methods do not explicitly model expert-desired C-C and C→Y relationships. The closest to our work is Causal CGMs (Dominici et al., 2025), while C²BMs (Felice et al., 2025) are also closely related. The core differences lie in modeling and implementing $G$. The former learns relationships between endogenous variables and their copies, and embeds the concept representations in a higher-dimensional space, but incur substantial additional computational cost and require optimizing multiple loss functions, which complicates training. C²BMs instead enforce an acyclic $G$ with linear relationships. Both approaches rely on a side-channel; however, neither explicitly accounts for the contribution of the side channel to the final prediction. Lastly, none of the prior works explicitly handle mutually exclusive concepts. Overall, our approach is more computationally efficient, with modular components and greater flexibility in the types of reasoning it supports.

**Concept Incompleteness.** CBMs rely on predefined concept sets, causing lower accuracy when the concept set is incomplete (i.e., not a sufficient statistic for the target) (Mahinpei et al., 2021; Yeh et al., 2020; Zarlenga et al., 2022). For instance, in Fig. 1, a garment labeled as "Tops" may correspond to multiple classes, making

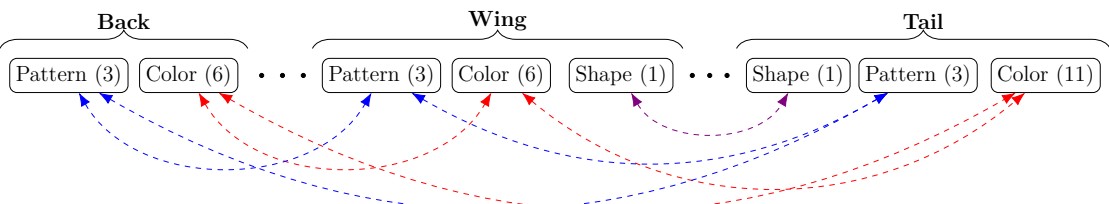

Figure 2: Partial illustration of CUB's reasoning graph. Concepts are represented as nodes, with numbers in parentheses indicating the cardinality for each concept. Nodes within the same group are mutually exclusive and disconnected, while edges are one-to-one between nodes from different groups. Bidirected edges indicate statistical dependencies between concepts.

exact classification impossible. Such cases arise when (i) concept annotation is costly, (ii) sparse explanations are preferred, or (iii) domain knowledge is limited. To address this issue, CBMs have been extended to incorporate side-channels. These hybrid CBMs have a lower upper bound of generalization error (Hayashi & Sawada, 2024) and capture unsupervised concepts (Sawada & Nakamura, 2022), residuals (Yuksekgonul et al., 2023; Zabounidis et al., 2023), or other auxiliary information (Dominici et al., 2025; Havasi et al., 2022). In contrast, we regularize the side-channel to prioritize concept importance.

**Concept Leakage.** Furthermore, CBMs suffer from concept leakage, a phenomenon tied to reasoning shortcuts (Bortolotti et al., 2024; 2025; Geirhos et al., 2020; Marconato et al., 2023b), where extra unintended information is encoded into concepts (Mahinpei et al., 2021; Makonnen et al., 2025; Marconato et al., 2023a; Margeloiu et al., 2021; Ragkousis & Parbhoo, 2024), leading to high accuracy even with irrelevant concepts and thus unreliable reasoning. Leakage has been suggested to be inherent in concept-based models that rely on concept embeddings (e.g., (Felice et al., 2025; Dominici et al., 2025; Zarlenga et al., 2022)) challenging their interpretability (Parisini et al., 2025). Existing methods to mitigate leakage include using binary concept representations (Havasi et al., 2022; Lockhart et al., 2022; Sun et al., 2024; Vandenhirtz et al., 2024), training a CBM model in an independent manner (Margeloiu et al., 2021), using orthogonality losses (Sheth & Ebrahimi Kahou, 2023) or disentanglement techniques (Marconato et al., 2022; Sinha et al., 2024). Meanwhile, the reasoning structure of CREAM allows only for intended information flows, thus mitigating leakage by design, without needing hard concepts or introducing additional regularization.

## 3 Concept Reasoning Model

In this work, we propose *reasonable* concept bottleneck models whose predictions are guided, but not strictly limited, by the designer-picked or automatically discovered `C-C` and `C→Y` relationships, while remaining practical, efficient and applicable even under incomplete concept sets. Unlike standard CBMs, which typically follow a bipartite concept-to-task architecture, we introduce CREAM as a framework that supports flexible, interpretable model reasoning while maintaining high performance.

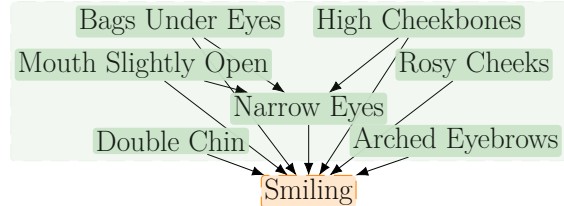

Figure 3: Reasoning graph for predicting "Smiling" in CelebA (Liu et al., 2015), showing the most correlated facial concepts ($C$) that *directly* influence the prediction ($Y$).

At the core of CREAM is the *model reasoning graph* $G = (V, E)$, which encodes the specified `C-C` and `C→Y` relationships. Formally, the node set is $V = C \cup Y$, and the edge set $E \subseteq V \times V$ captures plausible (un)directed relationships, restricting information flow within the model.

To operationalize this structure, we partition $G$ into two subgraphs: the concept graph[1] $G_C \coloneqq G[C] \triangleq (C, E_{G_C})$ containing the `C-C` relationships, and the task graph $G_Y \coloneqq G[C_{direct} \cup Y] \triangleq (C_{direct} \cup Y, E_{G_Y})$

---

[1]The graph can be disconnected, as seen in the seasonality concepts of cFMNIST, shown in Fig. 1.

for C→Y reasoning. Here, $C_{direct} := \{v \in C \mid \exists y \in Y : (v, y) \in E\}$ denotes the subset of concepts directly connected to $Y$.

## 3.1 Concept-Concept Reasoning

Although CREAM is not restricted to categorical concepts, we assume them in this context and represent them as one-hot encoded vectors of length equal to number of categories. For example, in the concept graph $G_C$ of Fig. 2, the concept "Tail Color" has cardinality 11 and is thus represented as a vector of 11 mutually exclusive binary variables. Henceforth, we assume the concept set $C$ is in this binarized form, with $K := |C|$ total concepts.

The concept graph's $G_C$ adjacency matrix, $A_C \in \{0, 1\}^{K \times K}$ is defined as follows:

$$A_C(i, j) = \begin{cases} 1 & \text{if } i = j \ \vee \ (c_i, c_j) \in E_{G_C}, \text{ for } c_i, c_j \in C; \\ 0 & \text{otherwise, i.e., undesired information flows.} \end{cases} \quad (1)$$

**Types of C-C Relationships.** $A_C$ captures both *hierarchical relationships* and *correlations* between them, represented as asymmetric ($A_C(i, j) \neq A_C(j, i)$) and symmetric ($A_C(i, j) = A_C(j, i)$) entries respectively.

For instance, in the graph of Fig. 3, all concept relationships are hierarchical (e.g., "High Cheekbones" lead to "Narrow Eyes" but not vice versa). In contrast, Fig. 2, includes bidirected edges accounting for concepts that are correlated (such as wing colors). By combining the above, $A_C$ can also represent a Partially Directed Acyclic Graph (PDAG). An example of this is shown in App. E.6. Lastly, in Table 1 we show how the different concept-based models and the relationships they encode, can be implemented within CREAM's framework.

Table 1: Implementation of prior approaches' C-C relationships in CREAM. For CGM we leave it at $A_C$, as it best describes a PDAG.

| Model | Relationships | $A_C$ |
|-------|---------------|-------|
| CBM | Independent | $\mathbb{I}_K$ |
| ACBM | Autoregressive | $(\mathbb{1}_{i<j})_{i,j=1}^K$ |
| SCBM | Correlations | $A_C = A_C^T$ |
| C2BM | DAG | sparse $(\mathbb{1}_{i<j})_{i,j=1}^K$ |
| CGM | Causal graph | $A_C$ |

## 3.2 Concept-Task Reasoning

Let $L := |Y|$ be the cardinality of the target variable $Y$. Following Section 3.1, in multiclass settings we assume the target variable has been binarized. We assume all C→Y edges are directed from concepts to task classes, reflecting experts reasoning "downwards" toward the targets.

Unlike prior, fully-connected CBMs, CREAM does not require all concepts to connect directly to the task. Indirect concepts, $C_{indirect} := C \setminus C_{direct}$ help predict other concepts within $G_C$, enhancing interpretability and enabling the intermediate steps of reasoning to be traced and verified. For instance, in Fig. 1, "Clothes" influences the final prediction indirectly through "Tops", which links to the target classes "T-Shirt", "Pullover", and "Shirt". Thus, the C→Y relationships lead to sparser explanations. For example, the class "T-Shirt" depends on just one concept rather than all of them.

As before, we encode the C→Y relationships in $G_Y$ using a task adjacency matrix[2] $A_Y \in \{0, 1\}^{K \times L}$

$$A_Y(i, j) = \begin{cases} 1 & \text{if } (c_i, y_j) \in E_{G_Y}, \text{ for } c_i \in C_{direct}, y_j \in Y; \\ 0 & \text{otherwise} \end{cases} \quad (2)$$

**Ease of Interventions.** Since only $C_{direct}$ is used for predictions, the total number of *effective interventions* is lowered from $K$ to $|C_{direct}|$. Users can also identify which concept predictions led to incorrect task predictions by tracing the edges of $G_Y$ and prioritize interventions on those specific concepts. Hence, $G_Y$ reduces human effort and complements existing concept selection criteria (Shin et al., 2023). Moreover, $G_C$ highlights relationships among concepts, such as mutually exclusive groups, which not only facilitates grouped interventions rather than individual ones (Koh et al., 2020; Shin et al., 2023) but also enables the

---

[2]Although adjacency matrices are conventionally square, we refer to this $K \times L$ binary matrix as an adjacency matrix for notational consistency.

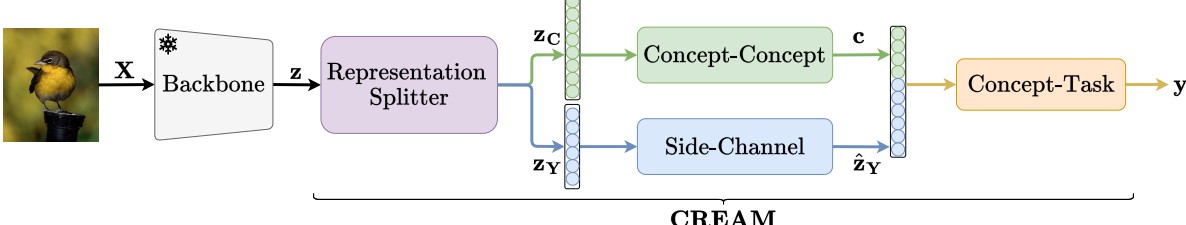

Figure 4: Sketch of CREAM's framework. The backbone's output is split into concept ($\mathbf{z}_C$) and side-channel ($\mathbf{z}_Y$) representations for concept and task prediction, respectively. The Concept-Concept block models relationships between concepts, while the Concept-Task block uses both $\hat{\mathbf{z}}_Y$ and the concepts to predict the task label based on embedded relationships.

propagation of interventions. By leveraging these concept relationships, changing one concept, such as the easier-to-intervene "Clothes," can automatically adjust downstream concepts like "Tops," ultimately affecting the task prediction, allowing $C_{indirect}$ to affect the task. Thus, given a known $G$, users can identify errors and correct them via interventions easier.

### 3.3 Concept Leakage

CBMs sometimes surpass the theoretical upper bound of concept-based task performance. This phenomenon, called *concept leakage*, is attributed to (i) concept representations inadvertently encoding extra information beyond their symbolic counterparts (Havasi et al., 2022; Mahinpei et al., 2021; Marconato et al., 2022; 2023a; Margeloiu et al., 2021; Sun et al., 2024; Parisini et al., 2025), and (ii) task predictions exploiting irrelevant concepts (Mahinpei et al., 2021; Sun et al., 2024). As a result, task predictions do not rely on learned concepts as intended, leading to erratic behavior under interventions and reduced interpretability. We define concept leakage as the surplus task accuracy of a model $f$ when using predicted concepts relative to the Bayes optimal predictor with true concepts: $\Lambda = \max(ACC_f - ACC_{Optimal}, 0)$ (Marconato et al., 2023a).[3] An exception occurs when models exploit side channels, since these can legitimately exceed the theoretical concept-based accuracy. Interventions on models with leakage may further reduce task accuracy, since human edits inject the exact intended information into the concepts (Margeloiu et al., 2021; Parisini et al., 2025). Our hypothesis, empirically validated in Table 4, is that enforcing a structured reasoning process through $A_C$ and $A_Y$ constrains spurious or semantically invalid pathways, compelling the model to use intended relationships, thereby reducing leakage and improving interpretability.

## 4 Designing CREAM

The CREAM framework, shown in Fig. 4, affords any CBM variant to be reformulated by embedding them with the following plug-and-play components: i) a *representation splitter* which decomposes the backbone feature representation ($\mathbf{z}$) into a concept representation ($\mathbf{z}_C$) and an optional *side-channel* representation ($\mathbf{z}_Y$); ii) a *concept-concept block* that enforces the `C-C` relationships via $A_C$; iii) a regularized side-channel that incorporates information not explicitly captured by the concepts; and iv) a *concept-task block* that encodes the `C→Y` reasoning via $A_Y$, and leverages the side-channel to improve task performance especially in concept incomplete cases. We showcase their modularity and effects in Section 5.2 and App. E.2, E.5.

### 4.1 Representation Splitter

CREAM builds atop a frozen pre-trained or fine-tuned backbone, which given an input image $X$ extracts a feature vector $\mathbf{z}$ serving as the initial information bottleneck. Then, a learnable *representation splitter* linearly partitions $\mathbf{z}$, into two disjoint latent representations:

---

[3]We adapt the definition in (Marconato et al., 2023a) from classification loss to accuracy.

1. **Concept exogenous variables** $\mathbf{z}_C \in \mathbb{R}^{d_C K}$ that serve as *input to the concept-concept block*, which enforces the C-C relationships. We assume a uniform latent capacity per concept for simplicity.

2. **Side-channel information** $\mathbf{z}_Y \in \mathbb{R}^{|\mathbf{z}|-d_C K}$ capturing information beyond the predefined concepts, that is necessary to fully predict the task.

The dimensionalities of both $\mathbf{z}_C$ and $\mathbf{z}_Y$, are hyperparameters that influence the model's performance. To incorporate the reasoning structure encoded by $G$ and the functional (in)dependence constraints it implies, we draw inspiration from Structural Causal Models (SCMs) (Pearl, 2009). In an SCM, each endogenous variable $X_i$ (here, $X_i \in \mathbf{X} = C \cup Y$) is modelled as a function of its causal parents $\mathrm{pa}\,(X_i)$ (given by C-C and C→Y) and an exogenous noise variable $\mathbf{z}_i \in \mathbf{z}_C \cup \hat{\mathbf{z}}_Y$, i.e., $X_i = f_i(\mathrm{pa}\,(X_i), \mathbf{z}_i)$.

**Structured Neural Networks.** Structured Neural Networks (StrNNs) (Chen et al., 2024) enforce the functional (in)dependence constraints implied by $G$, meaning a variable $X_i$ must not be influenced by the exogenous noise of a $X_j$ that is not its parent ($\frac{\partial X_i}{\partial \mathbf{z}_j} = 0, \{\forall j \mid X_j \notin \mathrm{pa}\,(X_i)\}$). Given the number of hidden layers $d$, layer widths $(h_1, h_2, \ldots, h_d)$, and an adjacency matrix $\mathbf{A} \in \{0,1\}^{p \times q}$ as hyperparameters, StrNN constructs a series of binary masks $M_1, \ldots, M_d$ which zero out non-permitted connections, ensuring the desired independencies are encoded while preserving maximal expressivity within those constraints. A detailed explanation of StrNNs can be found in App. A.

## 4.2 Concept-Concept Block

This block enforces the C-C relationships encoded in $A_C$. To improve predictive performance, w.l.o.g., we assume each concept is associated with a $d_C$-dimensional exogenous embedding, yielding an input representation $\mathbf{z}_C \in \mathbb{R}^{d_C K}, d_C \in \mathbb{N}$. To enforce the concept graph $G_C$, we generate binary masks using StrNNs: $M_C := A_C^T \otimes \mathbb{1}_{1 \times d_C}$, where $\otimes$ denotes the Kronecker product. This ensures that each concept receives input only from the appropriate parents' exogenous vectors. The resulting concept-concept block $g : \mathbb{R}^{d_C K} \to \mathbb{R}^K$ receives as input $\mathbf{z}_C$ and relies on the concept mask $M_C$ to compute the *concept logits*, $\hat{l}_C \in \mathbb{R}^K$ as :

$$\hat{l}_{C_i} = g(\mathbf{z}_{C_i}, \mathrm{pa}\,(\mathbf{z}_C)), \quad \text{where } \mathrm{pa}\,(C_i) = \{v \in V | A[v, C_i] = 1\}. \tag{3}$$

This formulation follows a compacted SCM principle (Javaloy et al., 2024), whereby each concept depends only on its parents and its corresponding exogenous variables. Standard CBM lacks this structure and connects all $\mathbf{z}_C$ to every concept allowing for entangled and unwanted reasoning paths. Note that we do not aim to be causally consistent (i.e., we do not impose any causal assumptions, nor explicitly define the type of equations $f$), but use causality as a guiding analogy.

**(Mutex) Concept Representations.** The concept-concept block supports hard, soft, logit, and embedding representations. For mutually exclusive *(mutex)* concepts in $G_C$, we apply a softmax over the logits within each group. For non-mutex concepts, we apply the respective activations independently. In the main paper we focus on soft concepts, i.e., $\hat{C} := \sigma(\hat{l}_C)$, where $\sigma$ is a sigmoid for non-mutex concepts and a group-wise softmax for mutex concepts, while an analysis of hard concepts in CREAM is found in App. G.

**Propagating Interventions in StrNNs** Our framework supports both standard interventions, which modify only the task prediction, and propagating interventions, which affect both concepts and the task. Since concepts are not directly connected, but only through their associated exogenous variables (Eq. 3), propagating an intervention requires reconstructing the activations of the layer immediately preceding the concept layer. This layer corresponds to the concept exogenous variables $\mathbf{z}_C$ when $d = 0$, and to the final hidden layer of the StrNN when $d > 0$. Once these activations are reconstructed, they are propagated forward through the network to obtain the updated concept values and the resulting task prediction (Fig. 5).

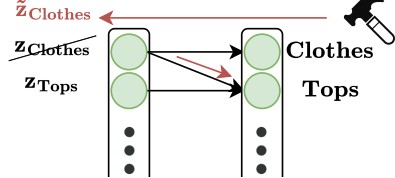

Figure 5: Illustration of the simplest case of propagating interventions in CREAM, corresponding to $d = 0$. For deeper graphs, the backward and forward passes is applied recursively.

To recover these activations, we invert the concept activations to recover their corresponding logits. For concepts in a mutex group, this inversion conditions on the remaining logits to correctly invert the softmax, whereas for sigmoid activations the standard inverse sigmoid is applied. Note that inversion is not possible for hard concepts due to the non-invertibility of the indicator function. Invertibility requires a one-to-one mapping between layers; i.e., the dimensionality of the preceding layer must match the number of concepts $K$ (e.g., when $d = 0$, each concept has a one-dimensional exogenous variable, i.e., $d_C = 1$). While a fully connected layer may be theoretically invertible, its inversion yields latent activations that jointly encode all downstream concepts. This prevents isolating the exogenous variable of a single concept without affecting the others. In contrast, StrNNs impose a sparse structural mapping in which each high-level concept has a unique parent. This structure enables the recovery of a single concept's exogenous variable, enabling interventions that do not interfere with unrelated concepts. Additional details are provided in App. A.3.

### 4.3 Side-Channel

The optional side-channel information $\mathbf{z}_Y$ is projected by an MLP to $\hat{\mathbf{z}}_Y \in \mathbb{R}^L$ and *serves as the exogenous input to the tasks*, assigning each class its own exogenous variable. W.l.o.g., we assume one exogenous variable per class, with each exogenous variable connected exclusively to its corresponding class. Increasing the dimensionality of $\hat{\mathbf{z}}_Y$ may improve performance but reduces concept importance. We empirically show in App. E.5.1 that the side-channel in CREAM primarily supports classes that cannot be predicted from concepts alone. Lastly, interventions do not affect the side-channel. When a concept is intervened upon, only the concept–task pathway is updated, while the side-channel input remains fixed at its predicted value.

**Regularization of side-channel.** Adding a black-box side-channel to CBMs boosts task performance, but may reduce interpretability, since task predictions can use non-interpretable predictors. To control this, we apply a dropout-based regularization (Huang et al., 2016), dropping the entire side-channel with probability $p$. This encourages the model to favor concepts, using the side-channel only when needed. At inference, the side-channel can be dropped for purely concept-based predictions. In practice, $p$ can be tuned via $CCI$ and a small grid search, requiring $CCI > 0.5$ (App. B.1.1), guided by a $C_{true} \rightarrow Y$ baseline as an indicator of concept completeness. We refer the reader to Section 5.2, and Section 6 for more information.

### 4.4 Concept-Task Classifier

The final stage of CREAM maps concept predictions to task logits while incorporating the side-channel in a controlled manner. Similar to the concept-concept block, we use StrNN to enforce C→Y relationships expressed by $A_Y$. To incorporate the side-channel representation $\mathbf{z}_Y$, we parameterize the concept-task StrNN using the binary mask $M_Y := [A_Y^T; I_L]$, where $I_L$ denotes the identity matrix of size $L$ that connects each element in the side-channel representation $\mathbf{z}_Y$ with only one of the tasks classes. This ensures each class is dependent only on its parent concepts (Eq. 3) and, the optional class-specific latent features from the side-channel, leading to *sparser explanations* compared to a CBM. Formally, the task prediction for class $j$, using a classifier $f$, usually a single layer MLP for interpretability, is computed as:

$$\hat{y}_j = f([\hat{c}_{Pa_j}, \hat{\mathbf{z}}_{Y_j}]) \quad \text{where } f : \mathbb{R}^{K+L} \rightarrow \mathbb{R}^L, \hat{c}_{Pa_j} \subseteq C_{direct}. \tag{4}$$

### 4.5 Training

To train CREAM, we adopt the joint bottleneck training scheme (Koh et al., 2020), which optimizes both the task loss ($\mathcal{L}_Y$) and the concept loss ($\mathcal{L}_C$) simultaneously, through linear scalarization. The optimization objective is to minimize the weighted sum of these losses ($\mathcal{L}$), for the observed training samples $\{(x^{(n)}, y^{(n)}, c^{(n)})\}_{n=1}^N$, where $x \in \mathbb{R}^{|\mathbf{z}|}$ is image embedding, $c \in \mathbb{R}^K$ are concepts, $y \in \{0,1\}^L$ is the target class, and $\lambda > 0$ is the weight of concept compared to task performance:

$$\mathcal{L} = \sum_n \mathcal{L}_Y(\hat{y}^{(n)}; y^{(n)}) + \lambda \sum_n \sum_k \mathcal{L}_{C_k}(\hat{c}^{(n)}; c^{(n)}). \tag{5}$$

# 5 Experiments

We evaluate our framework across standard datasets, demonstrating its ability to achieve the following desiderata. Our experiments assess: (i) concept and task accuracies, (ii) computational efficiency, (iii) intervenability, (iv) mitigation of concept leakage, and (v) the effect of the dropout regularization.

## 5.1 Setup

**Datasets.** We evaluate CREAM on three image datasets selected for their distinct relational structures. **FashionMNIST** (Xiao et al., 2017) exhibits hierarchical and mutex relations with concept incompleteness. We use two variants: Incomplete FMNIST (*iFMNIST*) with $K = 8$ hierarchical categories (Seo & Shin, 2019), and Complete FMNIST(*cFMNIST*) with $K = 11$ by adding seasonal attributes (Fig. 1). **CUB** (Wah et al., 2011; Koh et al., 2020) provides 112 correlated and mutex concepts describing fine-grained attributes such as tail color and wing pattern (Fig. 2). **CelebA** (Liu et al., 2015) involves a DAG structure over seven facial attributes used to predict smiling (Fig. 3). Full dataset details are found in App. C.

**Model Baselines.** To establish reference points for task performance, we train two baseline models: (i) a black-box model that uses the same backbone architecture as the corresponding CBM models, with a final classification layer appended for task prediction, and (ii) a model trained on the ground-truth concepts ($C_{true} \to Y$). For the latter, we use linear classifiers across all datasets. Further details on these models are provided in Appendix D. For hard concept representations, we include ACBM,[4] and SCBM, using only the Amortized SCBM since it consistently outperforms the Global SCBM. We also evaluate embedding-based models: namely $C^2$BM, $CGM_{CD}$ that discovers a graph and its version that embeds a given graph ($CGM_{prior}$). Finally, we include a CBM, and to showcase the modularity of our side-channel, a CBM with the side-channel (CBM+SC), as well as CREAM with and without it (CREAM w/o SC). Note that we tried including ECBM (Xu et al., 2024) which employs energy-based probabilistic modeling of `C-C`, `C-Y` relationships, but were unable to obtain satisfactory results with our backbone.

**Implementation Details.** All models are initialized from a shared backbone network, which is fine-tuned for a few epochs on the respective datasets and then frozen to ensure consistent feature extraction. For each dataset, we perform hyperparameter tuning; the selected configurations, detailed in App. D,[5] *use sufficiently high dropout rates to remain interpretable.* The construction of masking pathways in StrNNs follows the algorithm found in Zuko (Rozet et al., 2022). For all baseline models we adopt their proposed configurations.

## 5.2 Key Findings

**Bridging the gap between interpretability and performance.** The results in Table 2 show that CREAM achieves competitive task and concept performance, even under incomplete settings, outperforming both concept-based baselines and black-box models. Although CREAM's concept accuracy drops in CUB, its task performance remains competitive. Notably, CGM models are too slow for large graphs like CUB, due to their expensive graph operations, and $C^2$BM is incompatible with our imposed reasoning involving bi-directed edges on CUB.

The two main components of CREAM offer orthogonal benefits: structured reasoning promotes interpretability, while the side-channel addresses limitations of incomplete or noisy concepts. We observe that incorporating the side-channel consistently *improves both task and concept accuracy* across CBM and CREAM, supporting its effectiveness. We attribute the gains in concept accuracy primarily to improved optimization, as the model no longer needs to trade off concept accuracy against task performance, thereby reducing the likelihood of poor local optima. In App. F, we study the effect of different hyperparameters on CREAM's performance.

**Computational Efficiency.** We additionally evaluate the training computational efficiency of all models, reporting relative runtime and memory consumption compared to the standard CBM across datasets and runs. As CGM can only be executed on CPU [6], we restrict comparisons involving CGM to CPU settings,

---

[4]We use the implementation available in SCBM's repository.

[5]We will release our code publicly upon acceptance to ensure full reproducibility.

[6]CGM uses graph operations (strongly connected components and topological sorting) that do not support GPU execution.

Table 2: Task and concept accuracy (%). We report mean$_\text{std}$ over multiple runs. **Bold** indicates the best-performing method. Overall, CREAM achieves a strong balance between predictive performance and interpretability. See Table 1 for the reasoning structure associated with each model.

| Model | iFMNIST | | cFMNIST | | CUB | | CelebA | |
|---|---|---|---|---|---|---|---|---|
| | $ACC_Y$ | $ACC_C$ | $ACC_Y$ | $ACC_C$ | $ACC_Y$ | $ACC_C$ | $ACC_Y$ | $ACC_C$ |
| **Black-box** | $92.68_{0.00}$ | - | $92.68_{0.00}$ | - | $74.91_{0.02}$ | - | $78.47_{0.30}$ | - |
| $C_{true} \to Y$ | $60.00$ | - | $100.00$ | - | $100.00$ | - | $84.51$ | - |
| **CBM** | $91.19_{0.40}$ | $96.74_{0.36}$ | $92.00_{0.17}$ | $97.33_{0.06}$ | $73.76_{0.32}$ | $80.90_{0.08}$ | $79.28_{0.39}$ | $79.77_{0.17}$ |
| **ACBM** | $52.25_{5.18}$ | $98.94_{0.01}$ | $90.68_{0.09}$ | $98.03_{0.02}$ | $66.98_{0.43}$ | $\mathbf{94.15_{0.01}}$ | $81.40_{0.46}$ | $80.57_{0.53}$ |
| **SCBM** | $57.68_{0.63}$ | $98.86_{0.02}$ | $90.80_{0.17}$ | $97.54_{0.06}$ | $70.55_{0.19}$ | $90.28_{0.04}$ | $76.63_{0.47}$ | $80.54_{0.11}$ |
| **CREAM w/o SC** | $57.10_{0.05}$ | $\mathbf{99.07_{0.02}}$ | $\mathbf{92.31_{0.15}}$ | $97.52_{0.25}$ | $71.13_{0.22}$ | $85.66_{0.09}$ | $80.69_{1.13}$ | $78.46_{1.50}$ |
| **CBM+SC** | $91.02_{0.20}$ | $96.13_{0.20}$ | $92.18_{0.11}$ | $97.38_{0.08}$ | $\mathbf{74.36_{0.10}}$ | $82.79_{0.11}$ | $79.55_{0.24}$ | $79.97_{0.21}$ |
| **CGM$_{CD}$** | $68.81_{14.65}$ | $90.05_{5.28}$ | $67.92_{7.37}$ | $90.48_{1.20}$ | - | - | $\mathbf{81.70_{0.83}}$ | $\mathbf{82.17_{0.42}}$ |
| **CGM$_{prior}$** | $90.67_{0.21}$ | $98.77_{0.04}$ | $90.33_{0.19}$ | $97.72_{0.08}$ | - | - | $81.51_{1.36}$ | $81.99_{0.13}$ |
| **C$^2$BM** | $91.96_{0.18}$ | $98.96_{0.02}$ | $92.04_{0.23}$ | $\mathbf{98.07_{0.02}}$ | - | - | $76.19_{1.17}$ | $78.93_{0.58}$ |
| **CREAM** | $\mathbf{92.43_{0.23}}$ | $99.07_{0.03}$ | $92.38_{0.16}$ | $\mathbf{98.08_{0.06}}$ | $72.90_{0.28}$ | $86.83_{0.04}$ | $80.92_{0.55}$ | $79.91_{0.22}$ |

Table 3: Comparison of training time and peak GPU memory usage relative to CBM. For CGM, results are reported for CPU execution, along with overall system peak memory usage. All values indicate multiplicative factors relative to CBM (e.g., CREAM requires 1.82× the training time of CBM in iFMNIST).

| Model | iFMNIST | | cFMNIST | | CUB | | CelebA | |
|---|---|---|---|---|---|---|---|---|
| | Time | Peak Memory | Time | Peak Memory | Time | Peak Memory | Time | Peak Memory |
| **CBM+SC** | x$1.50_{0.14}$ | x$1.00_{0.00}$ | x$1.63_{0.21}$ | x$1.00_{0.00}$ | x$1.02_{0.03}$ | x$1.00_{0.00}$ | x$1.00_{0.00}$ | x$1.00_{0.00}$ |
| **ACBM** | x$5.30_{0.54}$ | x$1.04_{0.00}$ | x$7.90_{0.90}$ | x$1.04_{0.00}$ | x$14.13_{0.14}$ | x$1.25_{0.00}$ | x$3.27_{0.05}$ | x$1.07_{0.00}$ |
| **SCBM** | x$27.54_{2.82}$ | x$1.05_{0.00}$ | x$27.99_{3.72}$ | x$1.05_{0.00}$ | x$9.21_{0.26}$ | x$1.30_{0.00}$ | x$3.27_{0.05}$ | x$1.07_{0.00}$ |
| **CREAM w/o SC** | x$1.59_{0.10}$ | x$1.00_{0.00}$ | x$1.81_{0.24}$ | x$1.00_{0.00}$ | x$2.38_{0.12}$ | x$1.00_{0.00}$ | x$1.00_{0.01}$ | x$1.00_{0.00}$ |
| **CGM$_{CD}$** | x$9.15_{1.03}$ | x$361.87_{2.06}$ | x$20.24_{3.08}$ | x$1139.22_{3.01}$ | - | - | x$2.21_{0.07}$ | x$82.57_{2.61}$ |
| **CGM$_{prior}$** | x$8.47_{3.20}$ | x$36.14_{0.41}$ | x$8.47_{0.97}$ | x$40.63_{0.35}$ | - | - | x$2.23_{0.04}$ | x$18.64_{1.40}$ |
| **C$^2$BM** | x$12.19_{0.18}$ | x$1.00_{0.00}$ | x$15.14_{1.43}$ | x$1.00_{0.00}$ | - | - | x$2.85_{0.09}$ | x$1.94_{0.00}$ |
| **CREAM** | x$\mathbf{1.82_{0.16}}$ | x$1.00_{0.00}$ | x$\mathbf{1.94_{0.05}}$ | x$1.00_{0.00}$ | x$\mathbf{2.46_{0.21}}$ | x$1.00_{0.00}$ | x$1.00_{0.01}$ | x$1.00_{0.00}$ |

whereas all other models are evaluated on GPU. Our results show that, among all structural CBM variants with and without a side-channel, CREAM is the most computationally efficient model, achieving both the fastest runtime and the lowest memory usage, in all datasets. Incorporating the side-channel to CREAM or to CBM introduces only a negligible overhead. Additional experimental details and results are provided in App. E.1.

**Intervenability.** We also assess task accuracy after intervening. For models with hard concepts, interventions are straightforward: the true concept values are directly inserted. However, for soft concept models, we set the concept activations to the 5th and 95th percentiles proposed in (Koh et al., 2020). For all models, we follow a random concept selection policy (Shin et al., 2023). However, causal models differ in that they avoid intervening on both parent and child concepts in the graph. For CREAM we randomly select from $C_{direct}$, the only concepts used for prediction. This sets an upper bound on the number of interventions (6 for iFMNIST, 9 for cFMNIST), reaching peak accuracy faster. For comparison, we also intervene on $C_{indirect}$ for CREAM. In CUB and CelebA, all concepts are direct, thus the number of effective interventions is 112 and 7, respectively. In App. E.4 we also showcase experiments where we propagate interventions in CREAM.

Fig. 6 shows task accuracy across models after interventions. Overall, increasing the number of interventions generally improves performance across models and datasets; however, no model reaches $C_{true} \to Y$ accuracy in iFMNIST, due to the incomplete concept set. In iFMNIST, CBM performance decreases with more interventions, collapsing to the $C_{true} \to Y$ level, indicating a leaky model (Margeloiu et al., 2021). In CUB, both CBM and SCBM degrade under certain interventions, further highlighting their sensitivity to intervention noise. C2BM shows mixed behavior: in CelebA, its performance remains largely unchanged for the first few interventions and improves only after more than five interventions. CGM achieves strong performance, even surpassing $C_{true} \to Y$ in CelebA, albeit at the cost of interpretability. Specifically, CGM$_{CD}$ initially underperforms but improves substantially after intervening on most concepts. Notably, C2BM and CGM are not applicable to CUB. In contrast, CREAM (with a side-channel) demonstrates stable and

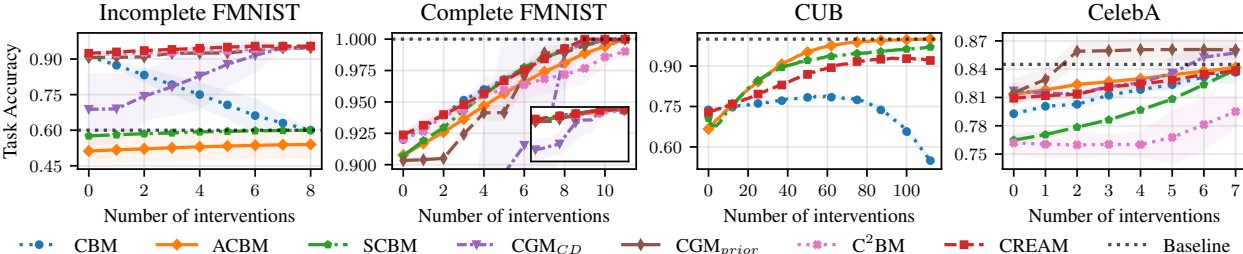

Figure 6: Impact of individual interventions on task accuracy. The baseline model is $C_{true} \rightarrow Y$. CREAM's accuracy improves with increasing number of interventions up to the number of $C_{direct}$. For cFMNIST, the inset axes show a zoomed-out view, to account for $CGM_{CD}$.

Table 4: CREAM's performance on iFMNIST without the side-channel. Leakage is avoided when using the C→Y relationships, while C–C relationships help mitigate it.

Figure 7: Absolute correlation matrix of the exogenous variables **z**. The enforced reasoning is mirrored in CREAM.

| Mutex | Reasoning | $ACC_Y$ | $ACC_C$ | $\Lambda$ |
|-------|-----------|---------|---------|-----------|
| No | C–C | $90.28_{4.2}$ | $96.38_{0.8}$ | $30.28_{4.2}$ |
| | C→Y | $57.31_{0.3}$ | $99.06_{0.0}$ | $0$ |
| | C–C, C→Y | $57.41_{0.6}$ | $99.04_{0.0}$ | $0$ |
| Yes | C–C | $67.60_{7.8}$ | $98.53_{1.1}$ | $7.60_{7.8}$ |
| | C→Y | $57.32_{0.2}$ | $99.05_{0.0}$ | $0$ |
| | C–C, C→Y | $57.10_{0.1}$ | $99.07_{0.0}$ | $0$ |

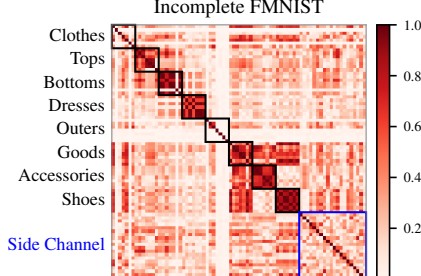

competitive performance, consistently benefiting from interventions and matching the baseline in CelebA after full interventions. Moreover, in iFMNIST and cFMNIST, CREAM reaches peak accuracy with fewer interventions, as only the subset of direct concepts is used for prediction, resulting in improved intervenability. Finally, we observe that group interventions on mutex concepts (App. E.3) and incorporating a side-channel (App. E.2) can further improve intervention effectiveness.

**Concept Leakage.** In Section 3.3 we defined concept leakage as $\Lambda = \max(ACC_f - ACC_{Optimal}, 0)$, i.e., the model should not outperform the $C_{true} \rightarrow Y$ baseline *when using only concepts*. We therefore focus on iFMNIST, the only dataset where non–side-channel models outperform the $C_{true} \rightarrow Y$ baseline. Also, soft-concept models are more prone to leakage in incomplete concept settings (Mahinpei et al., 2021; Parisini et al., 2025). As seen in Table 2, CBM exhibits concept leakage in iFMNIST, whereas CREAM without the side-channel does not.

By stripping away CREAM's plug-and-play components, we empirically show that its structured reasoning helps mitigate leakage. To isolate the mechanisms responsible, we *remove the side-channel* and then: (i) remove C–C reasoning (ii) remove C→Y reasoning, and (iii) replace softmax with sigmoid to treat concepts as independent. Table 4 shows that CREAM avoids leakage despite being a soft model. Specifically, C→Y reasoning entirely prevents it, while C–C relationships help mitigate it. Notably, softmax enforces mutual exclusivity, reducing leakage, whereas sigmoid allows for more leakage. Fig. 7 confirms that correlations among exogenous variables reflect the imposed relationships, e.g., {"Goods","Accessories","Shoes"} belong to the same sub-tree, are highly correlated. Moreover, the exogenous variables $\mathbf{z}_C$ associated with a given concept are strongly correlated with one another, yet largely uncorrelated with the exogenous variables of other concepts. This suggests that each multidimensional $\mathbf{z}_C$ encodes information specific to its respective concepts; the high intra-concept correlation indicates that its components capture the same underlying concept-specific signal, while the low inter-concept correlation reflects a structural disentanglement across concepts that is consistent with the imposed reasoning graph. Additional results are provided in App. E.5.

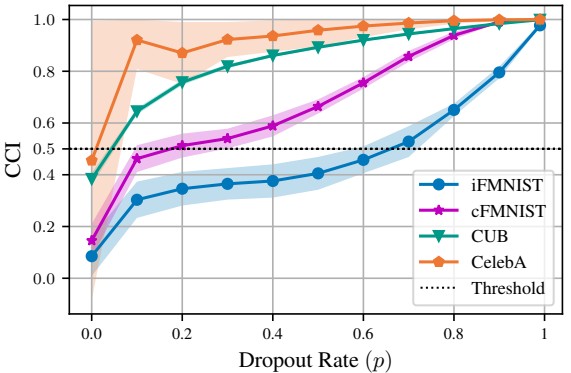 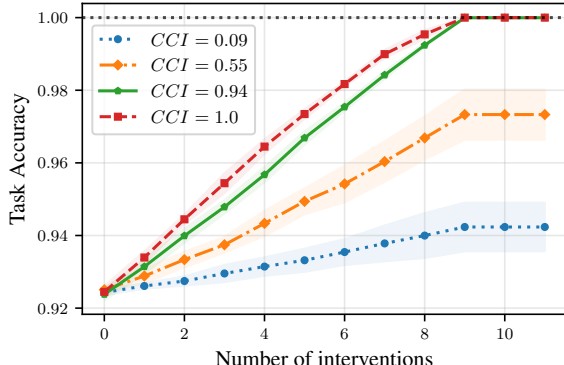

Figure 8: **Left:** Impact of dropout rates on concept channel importance. Models with complete concept sets need less dropout. **Right:** Interventions in CREAM on the cFMNIST dataset. Higher $CCI$ values, corresponding to higher dropout rates, reflect improved intervenability.

**Interpretability in the presence of a side-channel.** Because CREAM incorporates a black-box side-channel, it is crucial to verify that predictions primarily rely on the concepts. To this end, we introduce Concept Channel Importance ($CCI$), a metric adapted from SAGE values (Covert et al., 2020). SAGE extend Shapley values (Shapley et al., 1953) to measure *global feature importance* (Covert et al., 2020; Molnar, 2025) and, under an optimal model, corresponds to conditional mutual information.

We define $CCI$ as the normalized importance of the concept channel relative to total predictive capacity: $CCI = \frac{\phi_c}{\phi_c + \phi_y}$, where $\phi_c$ and $\phi_y$ denote the SAGE values of the concept and side channels, respectively. Values near 1 indicate stronger reliance on concepts, and thus higher interpretability. Importantly, computing SAGE (or Shapley) values in our setting is computationally efficient, as we treat the concept and side channels as two groups of variables, reducing the game to only two coalitions. Furthermore, since Shapley-based methods are model-agnostic, CCI is independent of the specific C→Y predictor used. Details on CCI and SAGE values as well as an analysis of our results based on permutation feature importance (Breiman, 2001; Fisher et al., 2019) can be found in App. B. In App. B.1.1, we show that $CCI > 0.5$ is enough for our desiderata to hold.

Fig. 8 (left) shows that increasing the dropout rate $p$ raises CCI, promoting concept-based reasoning. Also, the need for side-channel regularization decreases when using complete concept sets. Importantly, models that use almost zero regularization fall below the CCI threshold, highlighting the importance of side-channel regularization that prior works ignored. These findings lead to a key conclusion: *dropout rate can control interpretability in CBMs with side-channels.*

**Side-channel and interventions** Introducing a side channel can cause concepts to be ignored during task prediction. For example, when $CCI \approx 0$, the model ignores the concept predictions, so changing them will have no effect on the task output, i.e., interventions on the concepts become ineffective. Fig. 8 (right), illustrates interventions on CREAM for the cFMNIST dataset across different CCI values, each induced by a corresponding dropout rate. All models exhibit similar trends; as we intervene more, accuracy improves until it reaches the $|C_{direct}|$ bound. However, higher CCI values yield larger accuracy gains, as indicated by the steeper slopes of the accuracy–intervention curves. These results further support the intuition that *CCI indirectly measures intervenability.*

**Side-channel and (in)complete settings** Finally, we examine the effect of dropout on model performance. As shown Fig. 9 (right), increasing the dropout rate $p$ leads to a slight degradation in task performance in the complete settings. Surprisingly, even at extreme dropout levels $p \approx 1$, *CREAM retains black-box performance in incomplete datasets.* Furthermore, for fully concept-predictable classes (e.g., "Trouser", "Dress" in iFMNIST), the corresponding side-channel neurons exhibit nearly constant outputs. In App. E.5.1, we analyse the variance of these neurons in detail; for example, we observe substantially higher variance for concept-incomplete classes such as "T-shirt", "Shirt", and "Ankle Boot". Moreover, in App. E.5.1 we further show correlations between the side-channel exogenous variables and concepts, and observe no clear structure.

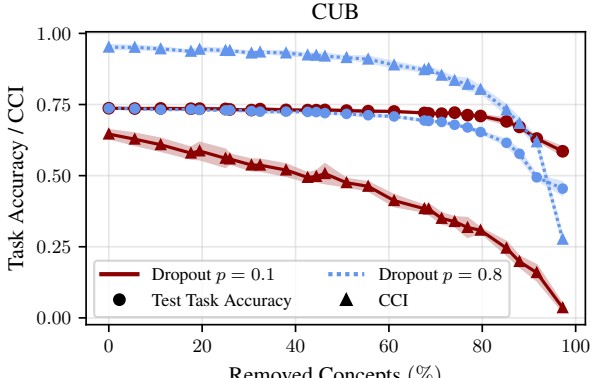 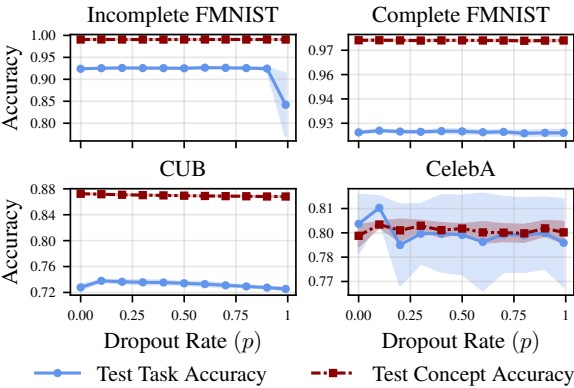

Figure 9: **Left:** Task accuracy and CCI for varying number of concepts in CUB. We lower the output size of the splitter, while keeping $d_C = 4$. **Right:** Concept and task accuracy vs dropout rate ($p$). Values are presented as mean $\pm$ standard deviation across 5 seeds. Increased $p$ may lead to drops in task accuracy.

We further investigate this behavior on the CUB dataset, focusing on performance under incomplete settings. In this experiment, concepts are removed group-wise, with entire mutex groups eliminated at a time. While the dimensionality of the exogenous features was reduced, we maintained the same multiplicity for both $d_C$ and the side-channel's $|\mathbf{z}_Y|$. All models were trained for the same number of epochs, and we tested two different dropout rates. As shown in Fig. 9 (left), CREAM maintains nearly the same accuracy even when up to 55% of the concepts are removed, demonstrating that the side-channel effectively compensates for missing concepts. As expected, reducing the number of concepts leads to a reduction in CCI. This decline is slower when $p = 0.8$, and the model remains interpretable even with only about 10% of the concepts. In contrast, models with $p = 0.1$ lose interpretability when roughly 40% of the concepts are removed. These results suggest that *as the number of available concepts decreases, stronger regularization of the side-channel becomes increasingly important.* **More broadly, our side-channel and regularization framework enables the effective use of CBMs, and thus interpretability in neural networks, even in regimes with very limited concept supervision.**

We hypothesize that as CCI decreases and accuracy is primarily driven by the side-channel, extra training epochs may be needed to fully exploit its capacity. This is because *dropout acts as regularization that slows the learning of the side-channel parameters*, helping to preserve interpretability while maintaining performance.

## 6   Practical considerations.

CREAM's modular design allows practitioners to make several choices, each with its own trade-offs, but all with high efficiency. We summarize trade-offs across configurations to help select models that best match priorities in interpretability, predictive performance, and intervenability.

The guiding principle is to select an efficient model that (i) matches the distributional structure of the concepts, (ii) minimizes concept leakage, and (iii) achieves the desired balance between performance and interpretability.

*Structural knowledge.* The reasoning graph can mirror the structure used by domain experts during annotation, or, when available beforehand, simplify annotation by constraining valid concept assignments. Incorporating prior knowledge also improves interpretability by grounding predictions in the specified structure. The more prior knowledge is available, the more it can be exploited: when fully known, it can be encoded directly as a reasoning graph; when only partially known, CREAM allows users to specify known relations while leaving others unconstrained (e.g., densely bidirectionally connected). Once defined, graphs can be reused across tasks, amortizing their cost. Moreover, because CREAM supports different reasoning graphs within the same architecture, practitioners can efficiently test and compare alternative structural hypotheses. Finally,

App. E.6 provides an example of automatically discovering a reasoning graph via causal discovery, reducing reliance on expert-designed structures.

*Concept representation and leakage.* Soft and hard variants (H-CREAM; App. G) apply to discrete concepts, while logits can be used for real-valued concepts. For mutually exclusive concepts, group-wise softmax should be used. In general, soft concepts tend to achieve better performance than hard concepts due to easier optimization and their ability to represent uncertainty. Also hard concepts, structured graphs, and softmax activations all reduce leakage (Table 4).

*Ease of interventions.* Models with explicit graph structure, such as CREAM, facilitate error tracing and enable more targeted interventions. Unlike vanilla CBMs, structured models can propagate interventions through learned or specified relationships. If practitioners specifically want interventions to propagate through the reasoning graph rather than affecting only the task prediction, the concept-concept block can be made invertible by setting $d_C = 1$ (App. A.3), at the cost of the modest accuracy gains provided by higher-dimensional exogenous representations (Fig. 19). When a side-channel is used, higher CCI values generally lead to stronger intervention effects (Fig. 8, right). Finally, hard concepts further simplify interventions by turning them into binary state changes rather than threshold-based heuristics (e.g. the 5th and 95th percentiles).

*Concept completeness and the side-channel.* When concepts do not fully determine the task, practitioners must either accept the resulting performance ceiling or add a regularized side-channel to recover the missing signal (CREAM, CBM+SC). Even when concepts are complete, the side-channel may still be useful under imperfect concept prediction due to noise or occlusion. The degree of side-channel regularization should reflect concept completeness, as it controls reliance on concepts: stronger regularization yields higher CCI, and higher CCI leads to more interpretable models and more effective interventions (Fig. 8). We recommend always maintaining $CCI > 0.5$. Concept completeness can be estimated by training a $C_{true} \to Y$ model, whose accuracy provides an upper bound given the observed concepts. A small gap to 100% accuracy suggests a nearly complete concept set and therefore less side-channel regularization is required to satisfy the $CCI > 0.5$ criterion, whereas a larger gap indicates greater concept incompleteness and the need for stronger regularization. The regularization strength can then be tuned by selecting the smallest value that satisfies $CCI > 0.5$ while maintaining strong task performance. In our experiments, $p \approx 0.8$ served as a strong default.

In summary, CREAM is a flexible framework that contains existing approaches (Table 1) and supports different reasoning graphs, concept representations, and side-channel configurations within a single model. This flexibility allows practitioners to encode knowledge in different forms, compare alternative specifications, and validate modeling assumptions within the same architecture. As a result, practitioners can systematically explore these trade-offs without switching model classes, while maintaining computational efficiency.

## 7 Conclusion and Future work

In this work, we introduced CREAM, a computationally efficient and flexible CBM framework that enables experts to encode prior knowledge about C–C and C→Y relationships into model reasoning, and explore multiple design choices without switching architectures. Its modular design supports diverse C–C relationships and concept representations, while the sparser C→Y reasoning facilitates interventions and interpretability. Empirically, we found that CREAM's structured reasoning effectively avoids concept leakage in specific scenarios, making it, to the best of our knowledge, the first soft CBM framework that is leakage-free.

Also, CREAM narrows the interpretability-performance gap, especially in concept-incomplete settings, through a regularized side-channel. In this regime, our approach enables CBMs to remain both effective and interpretable even when only a small number of concepts are available, significantly broadening their applicability to real-world settings with limited concept sets. To further evaluate models in this setting, we introduced $CCI$, a new model-agnostic metric for quantifying interpretability when predictions rely on auxiliary channels beyond the concept channel. Importantly, we showed that proper regularization of the side-channel (e.g., via dropout) is crucial to maintaining interpretability and intervenability in hybrid CBMs.

**Future Work and Limitations.** CREAM requires prior domain knowledge to encode `C-C` and `C→Y` relationships. In practice, this can be partially mitigated through graph reuse, partial specification, and automated structure learning (Zanga et al., 2022) (App. E.6). An additional promising direction is the use of Large Language Models (LLMs) to improve scalability. Recent work has shown that LLMs can enhance causal discovery pipelines by providing structural priors or refining discovered graphs (Kiciman et al., 2024; Ma, 2025). Separately, LLMs could also be used as standalone knowledge elicitation tools to directly infer `C-C` and `C→Y` relationships from textual domain knowledge, further reducing the need for manual graph specification. Improving the stability and reliability of intervention propagation via inversion presents an exciting avenue for future enhancement. Additionally, implementing adaptive and test-time dropout strategies, where the side-channel is dynamically leveraged based on concept prediction uncertainty, could improve robustness and interpretability while reducing the effort required for hyperparameter tuning. Furthermore, the structure learned in the exogenous variables and the side-channel can be leveraged to guide the discovery of new concepts (Sawada & Nakamura, 2022) that are not captured by the current concept bottleneck, for instance by identifying concepts that are semantically independent of, or hierarchically related to, existing concepts.

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

# A  Structured Neural Networks for Concept-Concept and Concept-Task Relationships

Structured Neural Networks (StrNN) (Chen et al., 2024) enforce functional independence between inputs and outputs using masking pathways that preserve the structural constraints dictated by an adjacency matrix. Given a function $f : \mathbb{R}^m \to \mathbb{R}^n$, where $z \in \mathbb{R}^m$ is the input and $\hat{z} \in \mathbb{R}^n$ is the predicted output, StrNN ensures that dependencies between inputs and outputs adhere to a given adjacency matrix $A \in \{0,1\}^{m \times n}$. This is enforced through the condition:

$$A_{ij} = 0 \implies \frac{\partial \hat{z}_j}{\partial z_i} = 0. \tag{6}$$

This means that if $A_{ij} = 0$, the output $\hat{z}_j$ remains independent of input $z_i$, maintaining the prescribed reasoning structure of the $G$.

**Masks in Structured Neural Networks**  For both concept and task prediction networks, layer-wise masking pathways are applied to enforce structured reasoning $A^{m \times n}$. Given a network with $d$ hidden layers, each with widths $h_1, h_2, \ldots, h_d$, we define binary masks:

$$M_1 \in \{0,1\}^{h_1 \times m}, \quad M_2 \in \{0,1\}^{h_2 \times h_1}, \quad \ldots, \quad M_d \in \{0,1\}^{n \times h_d}, \tag{7}$$

such that:

$$M' = M_d \cdot \ldots \cdot M_2 \cdot M_1 \approx M, \tag{8}$$

where $M' \in \{0,1\}^{n \times m}$ maintains the same sparsity pattern as $A^T$, ensuring structured dependencies are preserved. The structured neural network function $\text{StrNN}_M$ is defined as:

$$\hat{z} = \text{StrNN}_M(z) = f_{d+1}\left( \ldots f_1\left( (W_C \odot M_1) z + b_C \right) \right), \tag{9}$$

where each layer transformation follows:

$$f_i(z) = a\left( \left( W_{C(i)} \odot M_i \right) z + b_i \right), \quad \forall i \in \{1, \ldots, d\}, \tag{10}$$

where: $W_{C(i)} \in \mathbb{R}^{h_i \times h_{i-1}}$ is the learnable weight matrix at layer $i$, $M_i \in \{0,1\}^{h_i \times h_{i-1}}$ is the binary mask ensuring structured dependencies, $b_i \in \mathbb{R}^{h_i}$ is the bias term and $a(\cdot)$ is the activation function.

For the concept-concept learning block $\text{StrNN}_{M_C}$:

$$m = d_C K, \quad n = K, \quad h_{i \geq 1} = d_C K. \tag{11}$$

For the classifier $\text{StrNN}_{M_Y}$:

$$m = K + L, \quad n = L, \quad h_{i \geq 1} = K + L. \tag{12}$$

The depth $d$ is treated as a hyperparameter. By enforcing structured dependencies, StrNN ensures that both concept and task predictions follow expert-defined reasoning pathways, enhancing interpretability without sacrificing predictive performance.

## A.1  Concept-Concept Masking

The adjacency matrix for concept relationships is given by $A_C \in \{0,1\}^{K \times K}$, as defined in Section 3.1. The input to the Concept-Concept block is obtained from the representation splitter and is denoted as:

$$\mathbf{z}_C = (z_1, z_2, \ldots, z_{d_C K}) \in \mathbb{R}^{d_C K}. \tag{13}$$

Note that each dimension in $\mathbf{z}_C$ is a result of an expansion; each exogenous variable was duplicated from dimensionality of 1 to dimensionality of $d_C$. This operation is represented by the Kronecker product ($\otimes$). These extra dimensions must still follow the independencies of the original variable. Thus, we construct the concept mask $M_C \in \{0,1\}^{K \times d_C K}$ by:

$$M_C = A_C^T \otimes \mathbb{1}_{1 \times d_C}, \tag{14}$$

where $\mathbb{1}_{1 \times d_C}$ is a row vector of ones that replicates $A_C^T$ column-wise, ensuring structured reasoning of concept-concept relationships across all feature dimensions.

## A.2 Concept-Task Masking

For the concept-task classifier (Section 4.4), the input combines both concept predictions and the side-channel information. Thus, its input is the concatenated $\hat{C}$ and $\hat{\mathbf{z}}_Y$, and thus it is of size $\mathbb{R}^{K+L}$. The C→Y relationships are described in the adjacency matrix $A_Y \in \{0,1\}^{K \times L}$. Meanwhile, since we assign one dimension of side-channel variables ($\hat{\mathbf{z}}_Y$) to each task, we will need to expand the adjacency matrix used in StrNN, using an identity matrix $I_L$. From this, we can define the concept-task mask as:

$$M_Y = \left[ A_Y^T; I_L \right]. \tag{15}$$

This ensures that each class prediction depends only on the relevant parent concepts and its assigned side-channel node.

## A.3 Inversion and Interventions

**Benefit of StrNN**   One might argue that, in a fully connected layer, it is possible to recover the activations of the previous layer by inverting the weight matrix, assuming it is square and invertible. Formally, given:

$$C = W\mathbf{z}_C + b,$$

one could compute:

$$\mathbf{z}_C = W^{-1}(C - b).$$

However, this inversion does not isolate the exogenous contribution of a single concept. Instead, it yields a latent activation vector $\mathbf{z}_C$ that jointly explains all downstream concepts as they were activated. In other words, the recovered activation corresponds to a configuration that preserves the entire representation (i.e., activated concept values and intervention), rather than the exogenous value that would match intervention on a specific concept. This is due to the fact that in a fully connected layer each downstream concept depends on all upstream activations, a *many-to-one relationship*. Consequently, it is not possible to uniquely attribute or recover the exogenous variable associated with a single concept without simultaneously recovering the others. In contrast, StrNNs impose a sparse structural constraint in which each high-level concept is connected to a unique "parent" exogenous variable, a *one-to-one relationship*. This structural disentanglement enables recovery of the exogenous variable associated with an individual concept and supports targeted intervention propagation. Because dependencies are explicitly structured, intervening on one concept does not implicitly modify unrelated concepts. Therefore, while invertibility in fully connected layers allows reconstruction of joint activations, only the sparsity of StrNNs enable identification and propagation of concept-specific exogenous interventions.

**Inversion in CREAM**   Let $\tilde{C}_i$ denote the intervened value of concept $i$, where $i$ belongs to a mutex group $k$. Assuming the remaining logits $\hat{l}_{C_j}$ are known, the softmax formulation gives:

$$\tilde{C}_i = \frac{e^{\hat{l}_{C_i}}}{\sum_j e^{\hat{l}_{C_j}}} \iff e^{\hat{l}_{C_i}} = \tilde{C}_i \sum_j e^{\hat{l}_{C_j}} = \tilde{C}_i e^{\hat{l}_{C_i}} + \tilde{C}_i \sum_{j \neq i} e^{\hat{l}_{C_j}}$$

$$\iff (1 - \tilde{C}_i)e^{\hat{l}_{C_i}} = \tilde{C}_i \sum_{j \neq i} e^{\hat{l}_{C_j}}$$

$$\stackrel{\tilde{C}_i \neq 1}{\iff} e^{\hat{l}_{C_i}} = \frac{\tilde{C}_i}{(1 - \tilde{C}_i)} \sum_{j \neq i} e^{\hat{l}_{C_j}}$$

$$\iff \hat{l}_{C_i} = ln(\frac{\tilde{C}_i}{1 - \tilde{C}_i}) + ln(\sum_{j \neq i} e^{\hat{l}_{C_j}})$$

If $\tilde{C}_i = 1$, then necessarily $\sum_{j \neq i} e^{\hat{l}_{C_j}} = 0 \iff \hat{l}_{C_j} \to -\infty, \forall j \in \text{mutex}_k$. Thus, we assume that the intervened value $\tilde{C}_i \in [0, 1)$, which also aligns with the 5th and 95th percentile heuristic.

When mutually exclusive concepts are not considered, the inversion simplifies substantially, and the logits can be recovered directly as: $\hat{l}_{C_i} = \sigma^{-1}(\tilde{C}_i)$. Once the logits are recovered, we invert them to obtain the concept exogenous variables $\mathbf{z}_C$. This inversion is possible due to combination of the masking structure of StrNNs and the reasoning graph. For instance, in the iFMNIST dataset, the logit for the concept "Clothes" is given by $\hat{l}_{\text{Clothes}} = w\mathbf{z}_{\text{Clothes}} + b_{\text{Clothes}}$, where $w$ is a scalar, since we require $d_C = 1$. Keep in mind, that the weight $w \neq 0$, i.e., there must be an active connection between logit and concept, in order to be able to find the corresponding $\hat{l}_{\text{Clothes}}$ value.

However, inversion can introduce issues. For instance, in the softmax case, if a user deactivates a concept while the logits of the remaining concepts are sufficiently large, the resulting $\hat{l}$ may be negative, which in turn can yield $\mathbf{z}_{C_i} < 0$. This is incompatible with architectures that use ReLU activations, where $\mathbf{z}_C \in [0, +\infty)$. To address this issue, as well as the numerical instabilities arising when $\tilde{C}_i = 1$, we clip $\mathbf{z}_C$ to be non-negative and restrict $\tilde{C}_i \in [0, 1)$. Another solution that would be theoretically valid, is switching to an activation function with a range $(-\infty, +\infty)$, such as LeakyReLU.

## B Feature Importance Metrics

Given that CREAM integrates a side-channel that can contribute to task predictions, it is crucial to assess the relative importance of the concept set $C$ compared to it. To quantify this, we employ two model-agnostic metrics: Concept Channel Importance ($CCI$) and Permutation Feature Importance ($PFI$). These two metrics collectively provide an assessment of whether CREAM effectively balances interpretability and performance by ensuring that predictions remain grounded in human-understandable concepts rather than being dominated by the side-channel. In Table 5, we report the values of these importance metrics, for the models used in Section 5.

### B.1 Concept Channel Importance

Concept Channel Importance ($CCI$) is based on Shapley Additive Global Explanations (SAGE) (Covert et al., 2020), which provide model-agnostic feature importance scores. Specifically, the SAGE value $\phi_i(v_f)$ of feature $i$, represents the Shapley values (Shapley et al., 1953) for the cooperative game $v_f(S)$. The cooperative game $v_f$ represents the expectation of the per-instance reduction in risk when using a subset of features $S \subseteq D$:

$$v_f(S) = \mathbb{E}[\mathcal{L}(f_\varnothing(X_\varnothing), Y)] - \mathbb{E}[\mathcal{L}(f_S(X_S), Y)],$$

where $f_\varnothing(x_\varnothing)$ is the model prediction without using any features, (i.e., the mean model prediction $\mathbb{E}[f(X)]$), and $f_S(X_S)$ the prediction using only the subset of features $S$ of all features $D$ ($S \subseteq D$). In our case, $\mathcal{L}$ is given by Equation 5, $f$ is the concept-task classifier and $D = [\hat{C}, \mathbf{z}_Y]$. Given $v_f(S)$, SAGE is then calculated by:

$$\phi_i(v_f) = \frac{1}{|D|} \sum_{S \subseteq D \setminus \{i\}} \binom{|D| - 1}{|S|}^{-1} \Big( v_f(S \cup \{i\}) - v_f(S) \Big).$$

SAGE values provide global interpretability (Molnar, 2025) instead of explanations for individual predictions. The features that the model deems most useful will have positive SAGE values, non informative features have values close to zero, and harmful for the prediction features have negative values. Lastly, SAGE values can also measure a weighted average of conditional mutual information when they are used with an optimal model trained with the cross entropy or MSE loss. Specifically, the SAGE value of a feature $i$ used in optimal model $f^*$, is equal to:

$$\phi_i(v_{f^*}) = \frac{1}{|D|} \sum_{S \subseteq D \setminus \{i\}} \binom{|D| - 1}{|S|}^{-1} I(Y; X_i | X_S).$$

We also briefly mention some of the properties that SAGE satisfies:

- **Efficiency:** SAGE values sum up to the total predictive power of all of the features (SAGE value using all the features $D$):. $\sum_{i=1}^{K} \phi_i(v_f) = v_f(D)$.

- **Dummy:** If a feature makes zero contribution, i.e., if it is an uninformative feature, then $\phi_i(v_f) = 0$.

Interestingly, SAGE values can be generalized to group of features. Since our objective is to measure the overall contribution of the concept channel, we treat all of the concepts as one coalition. The SAGE value of a coalition represents how much this group of features improves the model's predictive ability.

Concept channel importance ($CCI$) can also be expressed is terms of total predictive power $v_f(D)$:

$$CCI = \frac{\phi_c(v_f)}{\phi_c(v_f) + \phi_y(v_f)} \overset{\text{Efficiency}}{=} \frac{\phi_c(v_f)}{v_f(D)}$$

where $\phi_c(v_f), \phi_y(v_f)$ denote the SAGE value of the whole concept-channel and side-channel respectively.

### B.1.1  Deriving the desired importance threshold

We desire the importance of the concept channel to be greater or equal to the importance of the side channel, i.e., $\phi_c(v_f) > \phi_y(v_f)$. Assuming that *both channels are informative* (i.e., $\phi_c(v_f), \phi_y(v_f) > 0$), we derive a desired lower threshold for the concept channel importance:

$$
\begin{aligned}
& \phi_c(v_f) \geq \phi_y(v_f) && \text{add } \phi_c(v_f) \text{ to both sides} \\
\iff\ & 2\phi_c(v_f) \geq \phi_y(v_f) + \phi_c(v_f) && \text{assuming } \phi_c(v_f), \phi_y(v_f) > 0 \\
\iff\ & \frac{1}{2\phi_c(v_f)} \leq \frac{1}{\phi_y(v_f) + \phi_c(v_f)} && \text{multiply both sides with } \phi_c(v_f) \\
\iff\ & \frac{\cancel{\phi_c(v_f)}}{2\cancel{\phi_c(v_f)}} \leq \frac{\phi_c(v_f)}{\phi_y(v_f) + \phi_c(v_f)} \\
\iff\ & \frac{1}{2} \leq CCI
\end{aligned}
$$

Meanwhile, if the side channel is uninformative, then due to the dummy property $\phi_y(v_f) = 0$, then $CCI = 1$. For the sake of completion, in the edge case where the side-channel makes the prediction less accurate, i.e., $\phi_y(v_f) < 0$, then $CCI \in (-\infty, 0) \cup (1, +\infty)$. Note that we do not notice such cases in our experiments. Lastly, $CCI = 0$, if the concepts are not used by the Concept-Task Block.

In conclusion, assuming that both SAGE values are positive, $CCI$ is bounded between $CCI \in [0, 1]$. A $CCI$ value close to 1 indicates that the model relies primarily on the concept channel, reinforcing interpretability. When $CCI \approx 0.5$, it suggests that both the concept and side-channels contribute equally to the predictions. To compute CCI, we evaluate the trained model on the entire test set, storing the classifier's inputs to estimate feature attributions.

### B.2  Permutation Feature Importance

Permutation Feature Importance ($PFI$) (Breiman, 2001; Fisher et al., 2019) measures the significance of a feature by evaluating how much the model's accuracy deteriorates when its values are randomly shuffled. A feature is considered important if permuting its values leads to a significant drop in accuracy, whereas an unimportant feature results in little to no change. The PFI score for a feature $j$ is computed as:

$$PFI_j = ACC_Y - \frac{1}{K} \sum_{k=1}^{K} ACC_{Y_{j,k}} \tag{16}$$

Table 5: Interpretability metrics: CREAM variants (S-CREAM for soft concepts, H-CREAM for hard concepts) show higher concept-channel importance relative to the side-channel's across all datasets. Also, S-CREAM exhibits larger CCI values compared to H-CREAM in FMNIST.

| Dataset | Model | $CCI \uparrow$ | $PCI \uparrow$ | $PSI \downarrow$ |
|---------|-------|----------------|----------------|------------------|
| iFMNIST | H-CREAM | $0.80_{0.01}$ | $0.66_{0.06}$ | $0.35_{0.00}$ |
|         | S-CREAM | $0.80_{0.02}$ | $0.59_{0.06}$ | $0.35_{0.00}$ |
| cFMNIST | H-CREAM | $0.88_{0.02}$ | $0.66_{0.03}$ | $0.07_{0.04}$ |
|         | S-CREAM | $0.94_{0.02}$ | $0.72_{0.03}$ | $0.03_{0.01}$ |
| CUB | S-CREAM | $0.96_{0.00}$ | $0.72_{0.00}$ | $0.01_{0.00}$ |
| CelebA | S-CREAM | $0.92_{0.11}$ | $0.30_{0.02}$ | $0.01_{0.01}$ |

where $ACC_Y$ is the test accuracy of the model, and $ACC_{Y_{j,k}}$ is the accuracy after randomly permuting the values of feature $j$ for the $k$-th iteration. In our case, we focus on evaluating the importance of entire channels (i.e., groups of features). Thus we permute all values within each channel (concept channel and side-channel) simultaneously. We denote concepts' feature importance as **P**ermutation **C**oncept **I**mportance (PCI) and the side-channel's as **P**ermutation **S**ide-Channel **I**mportance (PSI). We measure PFI on the test dataset (Molnar, 2025), and we permute for 100 iterations. Lastly, as mentioned, we desire the concept-channel to be more important than the side-channel, i.e., $PCI > PSI$. This inequality plays the same role as the importance threshold $CCI > 0.5$ derived in B.1.1. As seen in Table 5, the PFI-based metrics also indicate that CREAM prioritizes the concept-channel instead of the side-channel.

### B.3 Dropout Rate and PFI

In Fig. 10, we present the Permutation Feature Importance curves for all datasets and models , when trained with different dropout rates ($p$). For each $p$, we train a model with it, and then we plot its $PCI$ and $PFI$, showing how much does the test accuracy drop if we randomly permute the concept-concept and side-channel, respectively. As expected, the findings are consistent with the observed trends in Section 5.2. Increasing the dropout rate $p$ leads to increased PCI and decreased PSI. Also, the need for side-channel regularization decreases when using complete concept sets. The point where $PCI > PSI$, in iFMNIST, is around $p = 0.75$, but when moving to the complete FMNIST case it drops to around $p = 0.3$. One difference between the PFI metrics and CCI, is the lowest required dropout rate $p$ such that the side-channel stops dominating the concept-concept block (i.e., $CCI > 0.5$ or $PCI > PSI$). Across all datasets, the PFI-based approach suggests that we should regularize the side-channel more than CCI suggests.

## C Dataset Details

A summary of the datasets used in our experiments is provided in Table 6.

### C.1 Dataset Description

**Apparel Classification (FashionMNIST)** The FashionMNIST (FMNIST) [7] dataset (Xiao et al., 2017) consists of 70,000 grayscale images of clothing items across 10 classes. However, FMNIST does not provide predefined concept annotations or a dependency graph. To define concepts, we use high-level apparel categories derived from the hierarchical structure in (Seo & Shin, 2019). The resulting dependency graph $G$ forms a hierarchical tree, as shown in Figure 1.

We call this dataset Incomplete FMNIST. Since in hierarchical classification the classes at each level are mutually exclusive, the concepts of the same depth of the tree are also mutually exclusive. This means that the mutex concepts are grouped as such: {Clothes, Goods} and {Tops, Bottoms, Dresses, Outers, Accessories, Shoes}, leading to two groups of concepts. Note that, in the hierarchical classification setting the second group

---

[7] https://github.com/zalandoresearch/fashion-mnist, MIT License

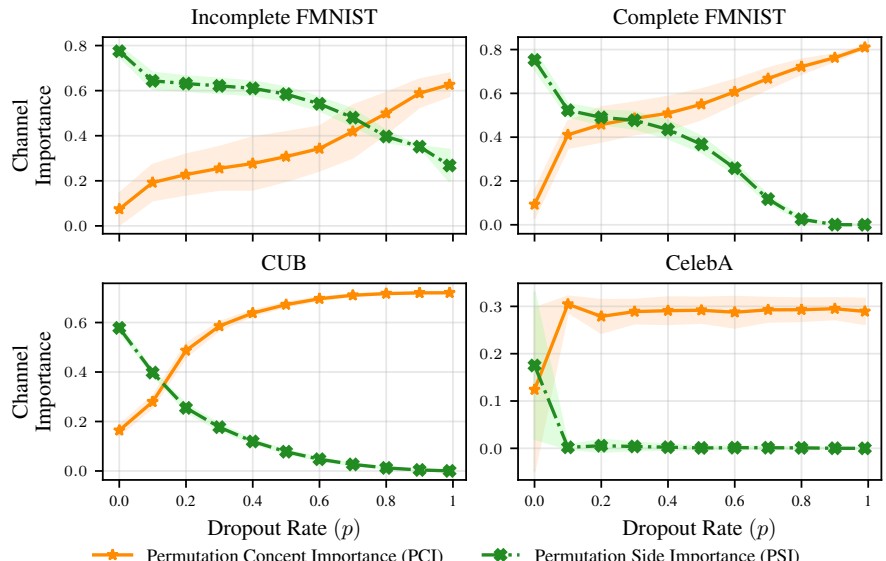

Figure 10: Mean values ± standard deviation of the permutation feature importance of both channels, in all datasets for CREAM, averaged over 5 seeds. Increasing $p$ leads to an increase in PCI and decreases PSI.

Table 6: Summary of each dataset. A concept group consists of a mutually exclusive group of concepts. $C_{direct}$ refers to the number of concepts directly connected to any task, which coincides with the upper bound of interventions. The reported number of effective group interventions refers to the number of directly connected groups of concepts instead.

| Dataset | Number of Classes ($L$) | Number of Concepts ($K$) | $C_{direct}$ | Mutex Groups | Effective Group Interventions |
|---|---|---|---|---|---|
| iFMNIST | 10 | 8 | 6 | 2 | 1 |
| cFMNIST | 10 | 11 | 9 | 3 | 2 |
| CUB | 200 | 112 | 112 | 27 | 27 |
| CelebA | 2 | 7 | 7 | 7 | 7 |

would be split into: {Tops, Bottoms, Dresses, Outers} and {Accessories, Shoes}. However, if we applied that logic to our case, both concept groups can be active at any time, thus being one mutex group. Lastly, these concepts cannot fully predict the classes. For instance, the same "active" concept vector $c = $ {Clothes, Tops} is used to predict these three classes $y = $ {T-shirt, Pullover, Shirt} with no way of distinguishing between them. The same problem applies for $c = $ {Goods, Shoes} and the classes $y = $ {Sandal, Sneaker, Ankle Boot}. This is where the side-channel shines; with the extra information from it, we can now successfully predict all classes.

For the Complete FMNIST dataset, we add a few more concepts to the hierarchical tree to make it a complete set. These concepts represent seasonality and are: {Summer, Winter, Mild Seasons}. We consider them to be mutually exclusive. The updated dependency graph is shown in Fig. 1.

**Bird identification (CUB)**   The Caltech-UCSD Birds-200-2011 (CUB) [8] dataset Wah et al. (2011) consists of 11,788 natural images spanning 200 bird species. Each image is annotated with 312 binary concepts describing visual characteristics such as color, shape, length, pattern, and size. Following (Koh et al., 2020), we process concept labels via majority voting across instance-level annotations and remove excessively sparse concepts, resulting in $K = 112$ concepts. These binary concepts originate from categorical attributes, leading to 27 mutually exclusive groups, similar to one-hot encoded features. To define concept relationships in $G_C$, we assume that concepts related to the same type of attribute (e.g., all colors, all patterns) are interconnected

---

[8]https://www.vision.caltech.edu/datasets/cub_200_2011/

via bidirected edges i.e. the concepts that pertain to the same "type" of feature are related, but we do not know its direction. These types of features are: {Shape, Color, Length, Pattern, Size}. Thus, the colors, patterns, etc. of all body parts are related. This structured representation embeds reasoning by enforcing relationships among semantically related features. Furthermore, we assume that all concepts directly influence all classes, ensuring that the model can fully leverage fine-grained feature representations.

**Smile Detection (CelebA)**   The CelebA [9] dataset (Liu et al., 2015) comprises over 200,000 celebrity face images annotated with 40 facial attributes. We select $K = 7$ facial attributes as concepts: ({Arched Eyebrows, Bags Under Eyes, Double Chin, Mouth Slightly Open, Narrow Eyes, High Cheekbones, Rosy Cheeks}), and use "Smiling" as the target label, making it a binary classification problem. The corresponding reasoning graph ($G$) is illustrated in Figure 3.

## D   Implementation Details

This section provides additional details on the implementation of our experiments. A more detailed version of CREAM's illustration is shown in Fig. 11. For datasets containing mutually exclusive concepts, we apply a softmax activation as described in Section 4. Our framework is implemented in PyTorch (v2.4.0) (Paszke et al., 2019) and PyTorch Lightning (v2.3.0). We use 20% of the stored activation values to perform the missing value imputation in CCI, and set SAGE's convergence threshold to $5 \times 10^{-2}$. The Permutation Feature Importance (PFI) metric is computed by permuting the channel values 100 times. All reported experiments use five different seeds. Lastly, for all ablation studies including the dropout rate, we use these $p$ values: {0.0001, 0.1, 0.2, 0.3, 0.4, 0.5, 0.6, 0.7, 0.8, 0.9, 0.99}.

**Details per Dataset**   Our experiments utilize three datasets, each with tailored architectures and preprocessing steps. We use the Adam optimizer (Kingma, 2014) for all models. For FashionMNIST (FMNIST), we employ a lightweight CNN backbone with two convolutional layers, ReLU activations, Max Pooling, dropout, and a final linear layer. The dataset is split into $50k - 5k - 10k$ for training, validation, and testing, respectively. The backbone is trained for 50 epochs, using a learning rate of $10^{-3}$ and a batch size of 256. Standard normalization is applied to the dataset. In the experiments of the main text, the number of epochs was also set to 50. Lastly, the standard model was trained from scratch for the same number of epochs.

For the rest of the datasets, we use ImageNet ResNet-18 (He et al., 2016) as the backbone, without the pretrained weights. CUB follows the same train-test splits, concept processing, and image preprocessing as in (Koh et al., 2020). The backbone is fine-tuned for 50 epochs, with a learning rate of $10^{-4}$ and a batch size of 64. For CelebA, we fine-tune ImageNet ResNet-18 on a 5K image subset for 90 epochs, using a learning rate of $10^{-4}$ and a batch size of 256, following (Yang et al., 2022). All three real-world datasets undergo identical preprocessing: color jittering, random resized cropping, horizontal flipping, and normalization. In the experiments of the main text, the number of epochs for CUB was 300, and in CelebA was 200.

**Model Selection**   In all cases, the side-channel consists of a Linear layer with a ReLU activation. We perform grid search over hyperparameters (Table 7). The Masked MLP depth refers to the number of hidden layers in the masked algorithm from Zuko (Rozet et al., 2022); a depth of zero means the Masked MLP is equivalent to a Masked Linear layer. Given that CBMs require multiple selection criteria, we adopt a ranking-based approach that averages performance across task and concept accuracies on the validation set, as in (Sanchez-Martin et al., 2024). Table 8 reports the best hyperparameter configurations for each dataset. Note that we eventually selected the models with the highest dropout rate in each case. Here, $d_C K + d_Y$ represents the total latent space dimension, while $d_Y$ refers to the side-channel input size. The depth (number of hidden layers) of the masked MLP is $d$. For the CBM model, in both iFMNIST and cFMNIST we used a learning rate of $10^{-2}$. For the $C_{true} \rightarrow Y$ model we trained linear classifiers for all datasets. Specifically, for iFMNIST and CelebA we tried a MLP with ReLUs and in total 3 layers, to try to improve the upper bound in task accuracy, without any significant improvements.

**Minimum Hardware Requirements**   All experiments were conducted on a high-performance computing cluster with automatic job scheduling, ensuring efficient resource allocation. We list the hardware requirements

---

[9] https://mmlab.ie.cuhk.edu.hk/projects/CelebA.html

Table 7: Hyperparameter search space explored for each dataset.

| Dataset | iFMNIST | cFMNIST | CUB | CelebA |
|---|---|---|---|---|
| Dropout $(p)$ | $\{0.2, 0.5, 0.8, 0.9\}$ | $\{0.2, 0.5, 0.8\}$ | $\{0.1, 0.2, 0.5, 0.8\}$ | $\{0.2, 0.5, 0.8\}$ |
| $d_C K + d_Y$ | $\{8, 18, 76, 78, 128\}$ | $\{11, 21, 22, 32, 43, 128\}$ | $\{424, 512, 648, 848\}$ | $\{7, 8, 10, 36, 70, 75, 256, 512\}$ |
| $d_Y$ | $\{0, 10, 20, 30, 64\}$ | $\{0, 10, 40, 62\}$ | $\{64, 176, 200, 400\}$ | $\{0, 1, 3, 5, 123, 162\}$ |
| $\lambda$ | 1 | 1 | 1 | $\{0.25, 0.75, 1\}$ |
| $d$ | $\{0, 2\}$ | $\{0, 2\}$ | $\{0, 3\}$ | $\{0, 5\}$ |

Table 8: Configurations for the best-performing models in each dataset. Note that $d_C K + d_Y$ represents the dimensionality of latent space that is split, and $d_Y$ the dimensionality of the input to the side-channel. H-CREAM and S-CREAM refers to the models with hard and soft concept representations respectively. The best models have $d = 0$ number of hidden layers in the Concept-Concept Block.

| Dataset | Model | $\lambda$ | lr | $p$ | $d_C K + d_Y$ | $d_Y$ |
|---|---|---|---|---|---|---|
| iFMNIST | H-CREAM | 1 | 1e-3 | 0.9 | 78 | 30 |
| | S-CREAM w/o SC | 1 | 1e-3 | - | 128 | 0 |
| | S-CREAM | 1 | 1e-3 | 0.9 | 76 | 20 |
| cFMNIST | H-CREAM | 1 | 1e-3 | 0.8 | 43 | 10 |
| | S-CREAM w/o SC | 1 | 1e-3 | - | 22 | 0 |
| | S-CREAM | 1 | 1e-3 | 0.8 | 128 | 40 |
| CUB | S-CREAM w/o SC | 1 | 1e-4 | - | 112 | 0 |
| | S-CREAM | 1 | 1e-4 | 0.8 | 648 | 200 |
| CelebA | S-CREAM w/o SC | 0.75 | 1e-3 | - | 7 | 0 |
| | S-CREAM | 1 | 1e-3 | 0.1 | 75 | 5 |

we used. For FMNIST (both iFMNIST and cFMNIST), we utilized 2 CPU workers, 8GB RAM, and a GPU with at least 4GB of VRAM. For CUB, the setup included 8 CPU workers, 16GB RAM, and a GPU with at least 4GB of VRAM. The CelebA experiments were more resource-intensive, requiring 12 CPU workers, 32GB RAM, and a GPU with at least 8GB of VRAM. The aforementioned resources refer to CBM and CREAM. A more detailed computational efficiency comparison can be found in App. E.1.

# E  Additional Results

## E.1  Efficiency

All experiments were conducted on a high-performance computing cluster with automatic job scheduling, ensuring efficient resource allocation. For the efficiency results we set specific hardware to ensure fair comparisons. For models using a GPU we set the hardware requirements to: 96GBs RAM, 8 cores out of an AMD EPYC 7662 64-Core Processor CPU, NVIDIA A100-PCIE-40GB GPU. For the CPU entries we set the hardware requirements to: 256GBs of RAM and we used all cores of a AMD EPYC 9654 96-Core Processor CPU. We compared exclusively the forward and backward pass of each model. To ensure a correct comparison by avoiding cache speedup, we used 5 burn-in iterations, and then for each model we report the mean values across 20 iterations. We also used 1 dataloader worker. We ensured correct timings and memory were recorded via using CUDA events, and tracemalloc in the CPU case. We also manually controlled the garbage collector.

We observe that across all datasets and devices, CREAM is relatively more computationally efficient than the rest of the models. Also, we notice that adding the side-channel to CBM (CBM+SC) only slightly decreases efficiency, supporting our claims about its efficiency.

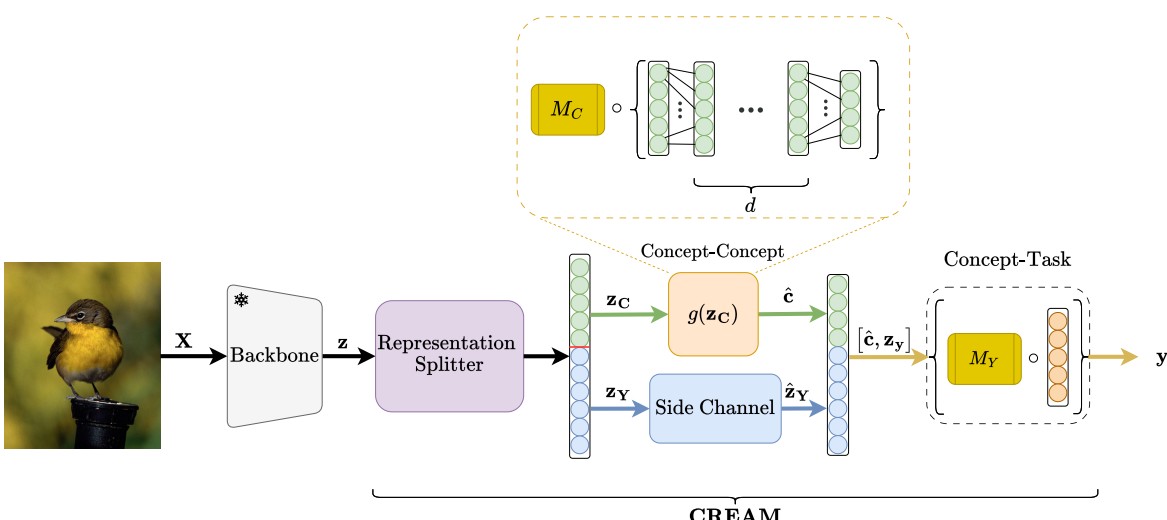

Figure 11: Expanded version of the illustration of CREAM found in the main text, providing additional details for clarity and completeness.

Table 9: Comparison of Training Time and GPU Peak Memory Usage relative to CBM. Values indicate the factor difference for models (e.g., CREAM takes $1.8\times$ longer to train than the baseline).

| Model | iFMNIST | | cFMNIST | | CUB | | CelebA | |
|---|---|---|---|---|---|---|---|---|
| | Time | Peak Memory | Time | Peak Memory | Time | Peak Memory | Time | Peak Memory |
| **CBM** | x$1.000_{0.000}$ | x$1.000_{0.000}$ | x$1.000_{0.000}$ | x$1.000_{0.000}$ | x$1.000_{0.000}$ | x$1.000_{0.000}$ | x$1.000_{0.000}$ | x$1.000_{0.000}$ |
| **CBM+SC** | x$1.496_{0.137}$ | x$1.001_{0.000}$ | x$1.631_{0.208}$ | x$1.001_{0.000}$ | x$1.017_{0.026}$ | x$1.004_{0.000}$ | x$1.004_{0.003}$ | x$1.001_{0.000}$ |
| **ACBM** | x$5.304_{0.535}$ | x$1.044_{0.000}$ | x$7.896_{0.900}$ | x$1.044_{0.000}$ | x$14.125_{0.140}$ | x$1.248_{0.000}$ | x$3.269_{0.050}$ | x$1.072_{0.000}$ |
| **SCBM** | x$27.535_{2.817}$ | x$1.050_{0.000}$ | x$27.985_{3.721}$ | x$1.053_{0.000}$ | x$9.210_{0.258}$ | x$1.303_{0.000}$ | x$3.269_{0.050}$ | x$1.072_{0.000}$ |
| **C$^2$BM** | x$12.194_{0.182}$ | x$1.003_{0.000}$ | x$15.143_{1.430}$ | x$1.005_{0.000}$ | - | - | x$2.846_{0.092}$ | x$1.938_{0.000}$ |
| **CREAM w/o SC** | x$1.585_{0.100}$ | x$1.001_{0.000}$ | x$1.810_{0.235}$ | x$1.000_{0.000}$ | x$2.377_{0.115}$ | x$1.000_{0.000}$ | x$1.002_{0.005}$ | x$1.000_{0.000}$ |
| **CREAM** | x$1.816_{0.159}$ | x$1.000_{0.000}$ | x$1.940_{0.048}$ | x$1.001_{0.000}$ | x$2.461_{0.212}$ | x$1.004_{0.000}$ | x$1.004_{0.006}$ | x$1.001_{0.000}$ |

Table 10: Comparison of Training Time (CPU) and overall system Peak Memory Usage relative to CBM. Values indicate the factor difference for models (e.g., CREAM takes $1.9\times$ longer to train than the baseline).

| Model | iFMNIST | | cFMNIST | | CelebA | |
|---|---|---|---|---|---|---|
| | **Time** | **Peak Memory** | **Time** | **Peak Memory** | **Time** | **Peak Memory** |
| **CBM** | x$1.000_{0.000}$ | x$1.000_{0.000}$ | x$1.000_{0.000}$ | x$1.000_{0.000}$ | x$1.000_{0.000}$ | x$1.000_{0.000}$ |
| **CGM$_{CD}$** | x$9.150_{1.033}$ | x$361.874_{2.061}$ | x$20.238_{3.084}$ | x$1139.224_{3.005}$ | x$2.209_{0.070}$ | x$82.571_{2.607}$ |
| **CGM$_{prior}$** | x$8.471_{3.195}$ | x$36.138_{0.405}$ | x$8.465_{0.969}$ | x$40.627_{0.351}$ | x$2.227_{0.035}$ | x$18.635_{1.404}$ |
| **CREAM** | x$5.387_{1.437}$ | x$1.259_{0.009}$ | x$6.609_{2.438}$ | x$1.259_{0.042}$ | x$1.963_{0.021}$ | x$1.122_{0.003}$ |

## E.2 Intervenability With and Without Side-Channel

In this section we study intervenability of CBM and CREAM with and without a side-channel. The results, visualized in Fig. 12 are similar to the ones in the main paper. However, we notice that including a side-channel seems to alleviate the drop in accuracy by intervening with a high number of interventions in CUB and CelebA.

## E.3 Group Interventions

In this section we examine the benefits of the `C-C` mutually exclusive relationships, on the efficiency of interventions. Specifically, we investigate group interventions, where contrary to individual interventions, the human expert intervenes on a group of mutually exclusive concepts simultaneously. For instance, in cFMNIST, a human expert would change the value of all concepts belonging in Mutex 3 ("Summer", "Winter" and "Mild

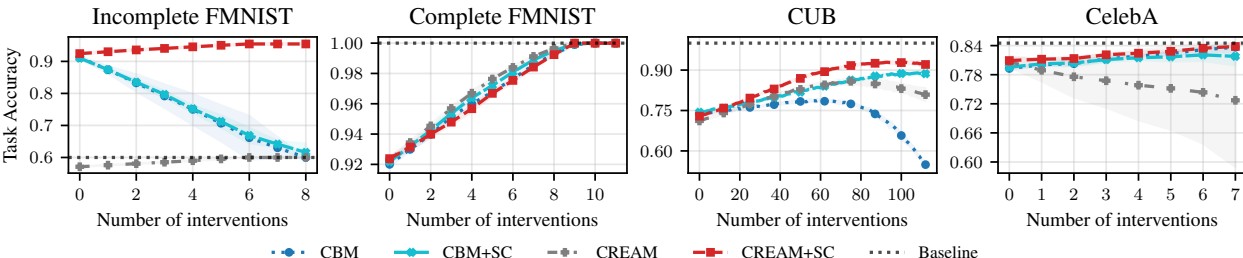

Figure 12: Impact of individual interventions on task accuracy. We reach similar conclusions as the ones in the main text.

Seasons") with just one intervention, by "activating" the concept "Summer" and the rest being deactivated automatically, since they are mutually exclusive. Note that, in the case of group interventions the upper bound of interventions drops even lower, down to the number of directly connected concept mutex groups. As visualized in Fig. 13, in FMNIST, we observe that task performance peaks after one group intervention in the incomplete setting and two group interventions in the complete setting. For both cases, the one remaining intervention, does not improve accuracy since those are the indirect mutex concepts ("Clothes" and "Goods"). Similar conclusions are seen in CUB; with about 20 group interventions we achieve performance gains of almost 90 individual interventions. Lastly, group interventions lead to the same performance gains as individual interventions, but with less human effort. These findings indicate that group interventions on mutually exclusive concepts, identified through C-C relationships, can help scale the intervention procedure.

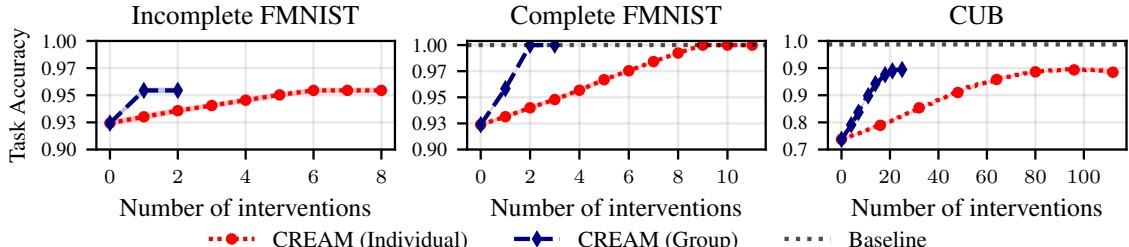

Figure 13: Impact of group interventions compared to individual ones, on task accuracy. Group interventions lead to the same performance gains with lower number of interventions. The number of relevant group interventions in CREAM for each dataset are, from left to right, 1, 2 and 27. Note the baseline's performance is not visualized in iFMNIST, as it is comparatively too low.

### E.4 Propagation of Interventions

We present experiments on both FMNIST variants in which interventions are propagated in CREAM using invertibility. Unlike the other intervention-based experiments, we focus here on intervening on $C_{indirect}$. The model hyperparameters follow those in Table 2, except that we set $d_C = 1$, to ensure invertibility. Additionally, because we account for mutually exclusive concepts, inversion is performed as in the softmax setting.

As shown in Fig. 14 intervening on $C_{indirect}$ results in almost no change in task accuracy; performance improves only once we begin intervening on $C_{direct}$. The same behavior is observed for CGM with a given graph on both iFMNIST and cFMNIST (Fig. 6). For CREAM, we attribute this effect in part to the inversion issues discussed in App. A.3. More generally, for both CGM and CREAM, this behavior can also be explained by the negligible influence of parent–child connections. Specifically, concept predictions appear to rely primarily on their own exogenous variables rather than on parent concepts in CGM, or on parents' exogenous variables in CREAM. This mirrors the issue observed between the side-channel and the concepts: in models with low CCI, intervening on the concepts similarly results in little to no change in task accuracy.

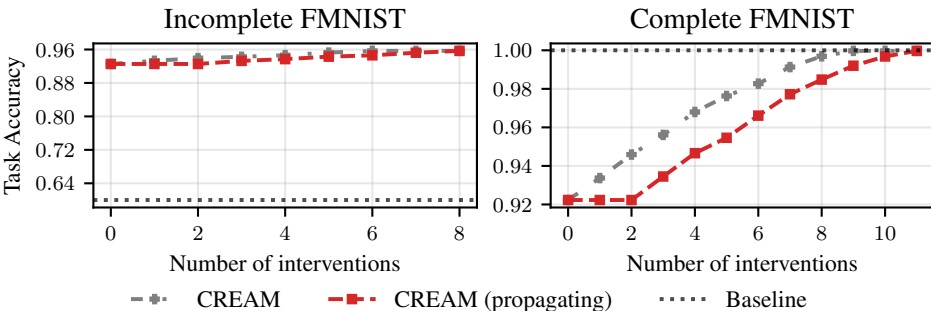

Figure 14: Propagating interventions via invertibility in CREAM. We intervene on $C_{indirect}$ first. In the other case, we intervene directly on $C_{direct}$.

Further investigation is needed to better understand and address these effects, which we leave for future work. For now, we recommend intervening directly on $C_{direct}$ to achieve maximal performance gains.

### E.5 Correlation of variables

In an effort to investigate the effect of different types of relationships on the model representations, we investigate their statistical relationships. Specifically, we visualize the absolute values of the correlations between the concept exogenous variables and the input to the side-channel. As seen in Fig. 15, in the FMNIST cases, where the C-C relationships create a DAG, the exogenous variables of the concepts form blocks in the correlation matrix. Each of the $d_C$ dimensions seem to be strongly correlated with the rest of the dimensions of the exogenous variable they belong to. Furthermore, we can observe the hierarchical structure of the concepts. For example, the exogenous of "Goods" is also strongly correlated to "Accessories" and "Shoes". Meanwhile, in CUB, where the C-C relationships create cycles in $G_C$, we do not observe any particular structure. In addition, CelebA exhibits a lot of variables with zero variance; they represent "dead" neurons. Lastly, in all cases, the side-channel also does not show a particular structure.

**Further investigating the block-diagonal structure** We will now focus on identifying the cause of the near block-diagonal structure in the correlation matrices. Following the leakage experiments from Table 4, we visualize the absolute correlations when the model reasoning is removed. According to Fig. 16, we identify C-C as the architectural choice leading to the near block-diagonal structure of the correlation matrix.

#### E.5.1 Variance of the side channel

Here, we will also check if some side-channel nodes have turned into constants, since we set the output of the side-channel to be of dimension $L$, which leads to an excess of needed nodes. For instance, in the iFMNIST dataset, the classes: {Trouser, Dress, Coat, Bag} are fully distinguishable from their assigned concepts, meaning that they do not need the side-channel. Thus, it would be preferable if their assigned side-channel nodes had a constant output. To verify this, we visualize the covariance matrix of the side-channel ($\hat{\mathbf{z}}_Y$), and we inspect its main diagonal. Specifically, for CelebA, $L = 1$ and thus there is only one side-channel node, we report here its variance: $Var_{\hat{\mathbf{z}}} = 0.0$, meaning that it is a constant value. As seen in Fig. 17, some of the nodes in iFMNIST (e.g., for "Trouser") and cFMNIST (e.g., for "Bag") have a low variance, meanwhile the classes that cannot be predicted only via concepts (e.g.,"T-shirt", "Shirt" , and "Ankle boot") have high variance, suggesting that CREAM primarily uses the side-channel information to predict only these classes.

### E.6 Causal Discovery and CREAM

One key distinction between CREAM and CGM Dominici et al. (2025) lies in how the reasoning graph is obtained. As noted previously, $\text{CGM}_{CD}$ employs causal discovery algorithms to infer the underlying causal (reasoning) structure from observational data. Thus, we will use one of these discovered graphs from $\text{CGM}_{CD}$ to simulate a causal discovery scenario. The discovered PDAG is visualized in Fig. 18. We will model, the

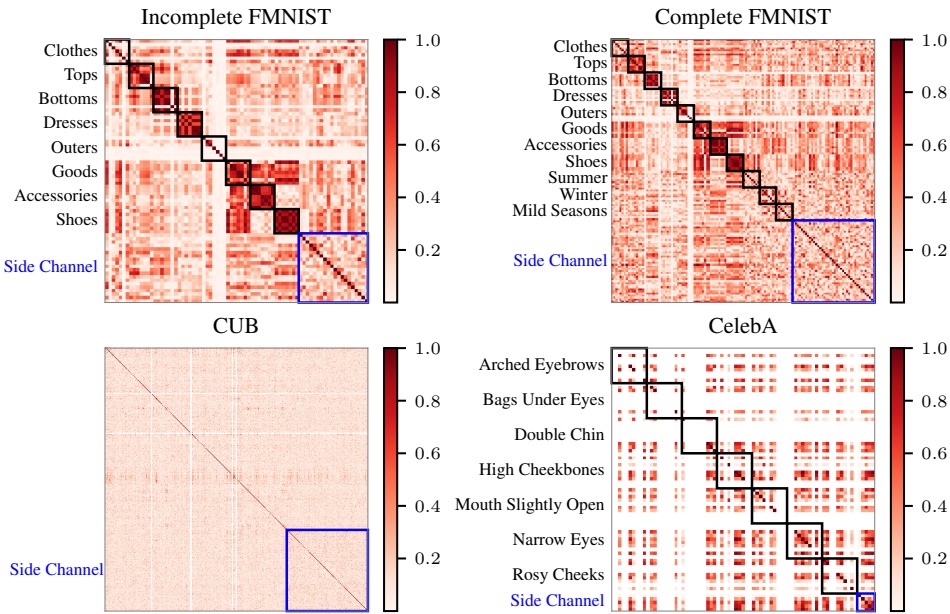

Figure 15: Absolute correlation matrix for the CREAM models reported in Table 4. For each concept we draw a box around its assigned $d_C$ dimensions. Note that CUB's exogenous are too small to plot. We notice that `C-C` reasoning leads block-like correlations, both within the exogenous of each concepts, and between their exogenous, revealing the hierarchical structure.

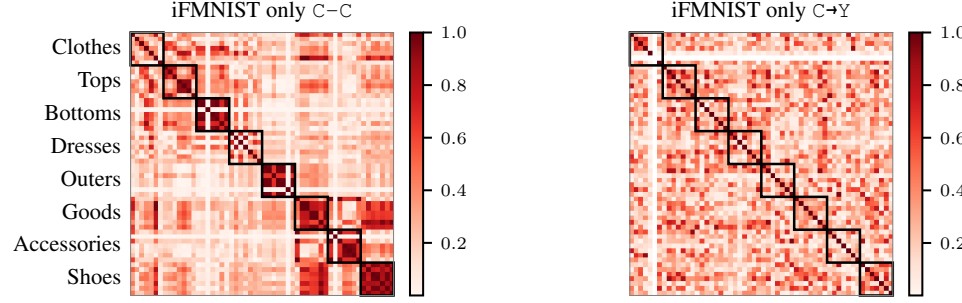

Figure 16: Absolute correlation matrix for the CREAM models reported in Table 4. For each concept we draw a box around its assigned $d_C$ dimensions. We notice that `C-C` reasoning leads block-like correlations, both within the exogenous of each concepts, and between their exogenous, revealing the hierarchical structure.

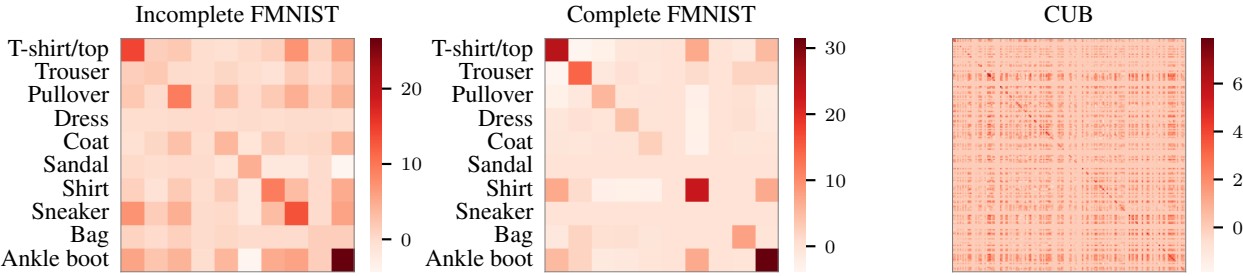

Figure 17: Covariance matrix of the outputs of the side-channel for CREAM. On the left, we report the corresponding class each node is assigned to. CUB has too many classes to clearly visualize.

undirected edges of the PDAG as symmetric entries in $A_C$, as we mentioned in Section 3.1. Using the same hyperparameters as the ones in the main text, we report: $ACC_Y = 79.56_{2.19}$, and $ACC_C = 79.55_{0.83}$. Thus, CREAM exhibits similar performance when using our DAG.

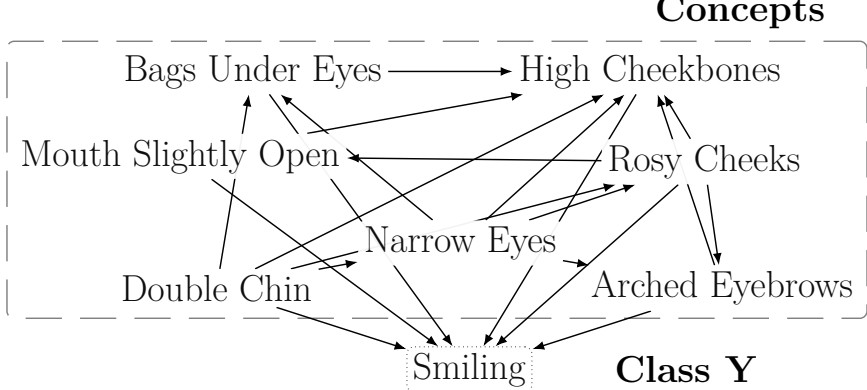

Figure 18: Reasoning graph taken from one the $CGM_{CD}$ runs Dominici et al. (2025) for predicting "Smiling" in CelebA (Liu et al., 2015).

## F  Ablation studies

In this section, we present additional experiments to offer a deeper understanding of CREAM's hyperparameters. We conduct ablation studies on multiple hyperparameters to illustrate their impact on different aspects of model performance, supplementing the main text with further insights.

### F.1  Effect of Dimensionality of Exogenous Variables

As mentioned in Section 4.2 each exogenous variable $\mathbf{z}_C$ is of dimension $d_C$. Here we study the effect of $d_C$ on model performance (as seen in Fig. 19), for $d_C \in \{1, 2, 3, 7, 10\}$. Note that, we keep every other hyperparameter to the same values as in Table 8. This means, we are increasing the dimensionality of the output of the representation splitter, while keeping $|\mathbf{z}_Y|$ constant.

We observe a general trend across all datasets; increasing the dimensionality $d_C$ of the exogenous variables of the concepts leads to same or slightly increased performance. That increase of performance is sometimes in the form of increased concept accuracy, task accuracy, or both at the same time.

### F.2  Effect of the depth of Concept-Concept Block

In Fig. 20 we study the effect of increasing the depth $d$ of the Concept-Concept Block to model performance. For each model we keep the same configuration as the main text, but now $d \in \{0, 1, 2, 3, 5\}$. We notice a trend across all models; increasing the depth leads to equal or worse performance. These findings further support the notion that human-interpretable concepts are often linearly encoded in the latent space of neural networks (Rajendran et al., 2024).

## G  Hard models

In this section we will study CREAM with hard concept representations, i.e., $\hat{C} \in \{0, 1\}$. To ensure trainability in the hard case, we leverage the straight-through estimator (STE) (Bengio et al., 2013), which approximates gradients during the backward pass. The representation of hard and soft concepts is given as:

$$\hat{C} := \begin{cases} \sigma(\hat{l}_C) & \text{S-CREAM} \\ round(\sigma(\hat{l}_C)) & \text{H-CREAM} \end{cases} \tag{17}$$

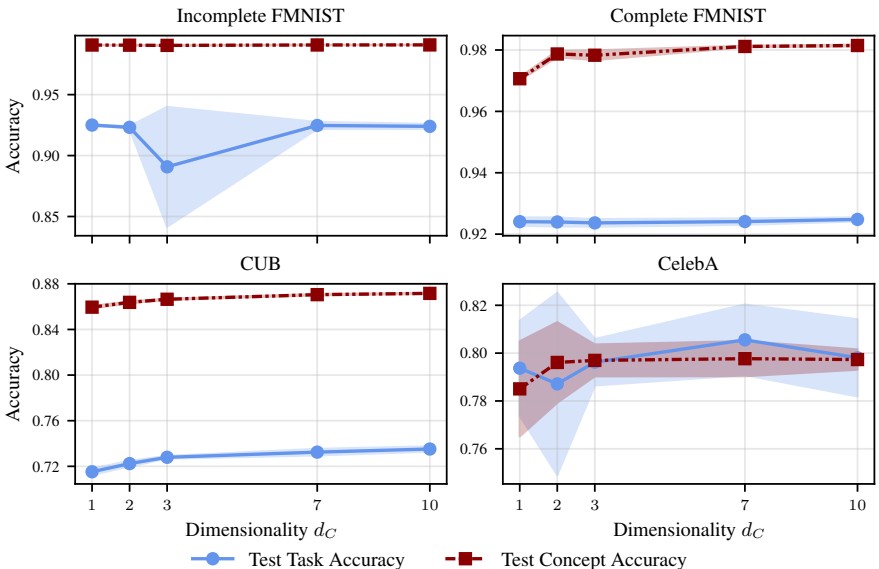

Figure 19: Concept and Task accuracies across different $d_C$ values for the exogenous of the concepts. Values are presented as mean $\pm$ standard deviation. Increasing $d_C$, generally improves performance.

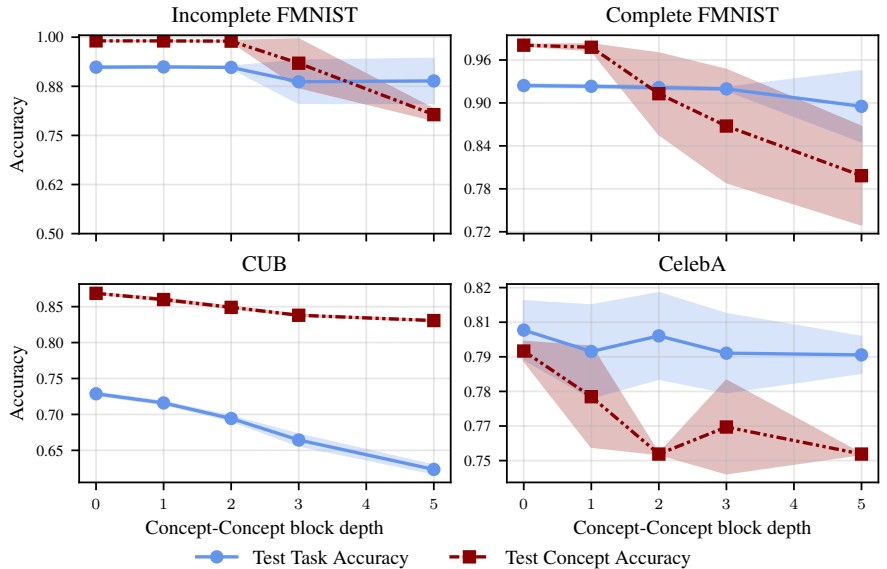

Figure 20: Concept and Task accuracies across different $d$ values in the Concept-Concept block. Values are presented as mean $\pm$ standard deviation. Increasing $d$ of the `C-C` block degrades performance.

where H-CREAM and S-CREAM are for hard and soft concepts respectively. We also handle mutually exclusive concepts similarly. The hyperparameter configurations of H-CREAM, can be found in Table 8. Hard CBM models are sometimes preferred due to them not suffering from leakage (Havasi et al., 2022; Vandenhirtz et al., 2024) and being easier to intervene on.

## G.1 Concept and Task Accuracy

S-CREAM consistently outperforms its hard variant across all datasets and evaluation metrics. This suggests that soft concept representations provide greater flexibility and are easier optimize, allowing the model to capture useful variations in concept values while maintaining structured reasoning.

Table 11: Performance comparison between CREAM variants. Reported values represent the mean and standard deviation over five seeds. S-CREAM outperforms H-CREAM performance and interpretability in both FMNIST variants. The best-performing model variant is highlighted in **bold**.

| Dataset | Model | Test $ACC_Y$ | Test $ACC_C$ |
|---------|-------|--------------|--------------|
| iFMNIST | S-CREAM | $\mathbf{92.43}_{0.23}$ | $\mathbf{99.07}_{0.03}$ |
|         | H-CREAM | $92.15_{0.17}$ | $98.98_{0.01}$ |
| cFMNIST | S-CREAM | $\mathbf{92.38}_{0.16}$ | $\mathbf{98.08}_{0.06}$ |
|         | H-CREAM | $91.29_{0.73}$ | $96.70_{0.17}$ |

### G.2 Importances

Table 5 demonstrates that, CCI remains consistently above the critical 0.5 threshold, and PCI exceeds PSI, for the selected hyperparameter settings. This indicates that the hard variant can also fulfill our concept importance desiderata. Furthermore, soft models report a higher CCI value compared to their hard counterparts, indicating that their decisions are more strongly influenced by the concepts.

### G.3 Interventions

We compare task accuracy between the two CREAM variants, after individual concept interventions. We follow the same process as the one mentioned in Section 5. For H-CREAM we do not have to find the 5th and 95th percentiles, and we instead directly use the ground truth concepts. As seen in Fig. 21, H-CREAM shows the same behavior as S-CREAM. However, the latter slightly outperforms the former in both presented cases.

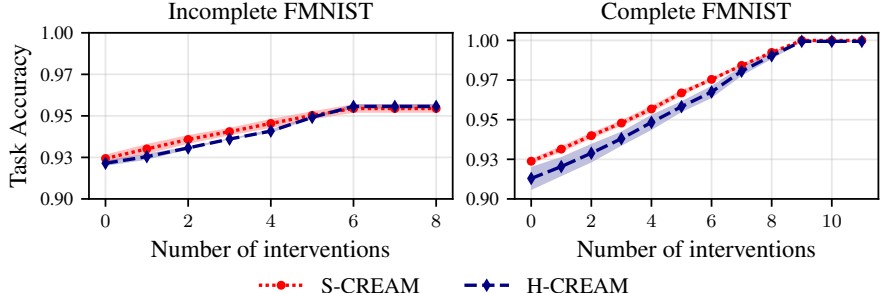

Figure 21: Intervenability comparison of H-CREAM and S-CREAM. The latter slightly outperforms the latter. Both models' task accuracy peaks after 6 and 9 interventions in iFMNIST and cFMNIST respectively.

### G.4 Correlations

We also investigate the correlations between the concept exogenous variables and the input to the side-channel in the hard variant. As illustrated in Fig. 22, H-CREAM does not exhibit the block correlations that the soft model does (Fig. 15). Also, in H-CREAM more concept exogenous variables have become constants. This indicates that the hard variant requires a lower number of exogenous variables, which is also in line with the best hyperparameters seen in Table 8.

## H   Logic viewpoint of CREAM

In this section we will express the relationships in $G$ using description logic (Baader, 2003). Note that in CREAM the concepts are calculated using the exogenous variables, and the tasks using the concepts and their corresponding side-channel input. Table 12 shows some of the logic rules that match CREAM's calculations, with a slight abuse of notation. For instance, a mutex constraint is described as: Clothes $\sqcap$ Goods $\sqsubseteq \bot$, and concepts are calculated by: Tops $\leftarrow \mathbf{z}_{Clothes} \sqcap \mathbf{z}_{Tops}$.

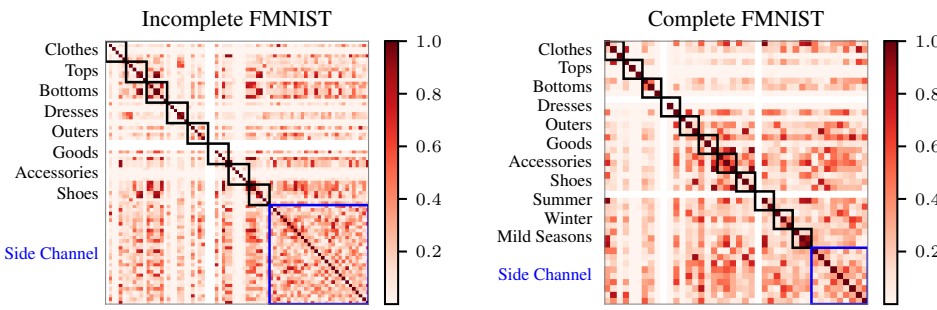

Figure 22: Absolute correlation matrices of the output of the representation splitter ($\mathbf{z}$) for H-CREAM. We notice that more concept exogenous variables have lowered correlations compared to the soft models.

Table 12: Some logic rules based on description logic for CREAM. The $\mathbf{z}$ variables correspond to the concept and side-channel, and $\perp$ denotes the empty set.

| Dataset | |
|---|---|
| iFMNIST | Clothes $\leftarrow \mathbf{z}_{Clothes}$ 
 Goods $\leftarrow \mathbf{z}_{Goods}$ 
 Clothes $\sqcap$ Goods $\sqsubseteq \perp$ 
 Tops $\sqcap$ Bottoms $\sqsubseteq \perp$ 
 Tops $\sqcap$ Dresses $\sqsubseteq \perp$ 
 $\vdots$ 
 Accessories $\sqcap$ Shoes $\sqsubseteq \perp$ 
 Tops $\leftarrow \mathbf{z}_{Clothes} \sqcap \mathbf{z}_{Tops}$ 
 $\vdots$ 
 Shoes $\leftarrow \mathbf{z}_{Goods} \sqcap \mathbf{z}_{Shoes}$ 
 T-shirt $\sqcap$ Pullover $\sqsubseteq \perp$ 
 T-shirt $\sqcap$ Shirt $\sqsubseteq \perp$ 
 $\vdots$ 
 Sneaker $\sqcap$ Ankle Boot $\sqsubseteq \perp$ 
 T-shirt $\leftarrow$ Tops $\sqcap \mathbf{z}_{T-shirt}$ 
 $\vdots$ 
 Ankle Boot $\leftarrow$ Shoes $\sqcap \mathbf{z}_{AnkleBoot}$ |
| CelebA | Bags Under Eyes (BUE) $\leftarrow \mathbf{z}_{BUE}$ 
 High Cheekbones (HC) $\leftarrow \mathbf{z}_{HC}$ 
 Mouth Slightly Open (MSO) $\leftarrow \mathbf{z}_{MSO}$ 
 Rosy Cheeks (RC) $\leftarrow \mathbf{z}_{RC}$ 
 Arched Eyebrows (AE) $\leftarrow \mathbf{z}_{AE}$ 
 Double Chin (DC) $\leftarrow \mathbf{z}_{DC}$ 
 Narrow Eyes (NE) $\leftarrow \mathbf{z}_{MSO} \sqcap \mathbf{z}_{BUE} \sqcap \mathbf{z}_{HC} \sqcap \mathbf{z}_{NE}$ 
 Smiling $\leftarrow BUE \sqcap HC \sqcap MSO \sqcap NE \sqcap RC \sqcap DC \sqcap AE \sqcap z_{Smiling}$ |

