# OpenReview forum: "Towards Reasonable Concept Bottleneck Models"
_TMLR — Under review for TMLR_

### Review · Reviewer_guQj · 2026-04-24

**Summary Of Contributions:**

The manuscript proposes a causal CGM architecture called CREAM similar to [1]. The only difference to prior work is that the proposed method has a side channel that aims at capturing all information that is not captured by the concepts. The method is empirically evaluated on FashionMNIST, CelebA and CUB. In addition, an agnostic metric that quantifies interpretability when predictions partially rely on the side-channel is introduced. The experiments show that CREAM enables the effective use of CBMs in regimes with very limited concept supervision.

[1] Dominici, Gabriele, et al. "Causal concept graph models: Beyond causal opacity in deep learning." International Conference on Learning Representations (2025).

**Audience:**

Yes

**Audience Explanation:**

I am not an expert at these concept bottleneck models, but perhaps someone will be interested to know that performance can be further improved using this side channel. It is a somewhat obvious insight, but that does not matter for this venue.

**Claims And Evidence:**

No

**Claims Explanation:**

From what I know, a major reason in favor of these CGMs is their interpretability. With this side channel, that argument is somewhat obsolete. It is written in Section 5.2 that *"it is crucial to verify that predictions primarily rely on the concepts"*, but in practice it is hard to quantify to what extend this is happening and it also questions the motivation of the side channel to begin with. I have some soundness concerns here and would ask the Authors to clarify this.

**Requested Changes:**

# General

- The title is quite poor, in my opinion. What does "reasonable" even mean? I strongly recommend modification of the title.

- Another aspect is that the method is not really a concept *bottleneck* model anymore, because the side channel obviously alleviates the bottleneck. I suggest discussing this aspect in greater detail and adjusting the title and narrative accordingly.

- Maybe I missed it, but how do interventions propagate into the side channel? I suggest clarification.

## Figure 1

- I suggest explicitly mentioning the $C-C$ and $C \mapsto Y$ relationships in Figure 1. It will make the manuscript more accessible.

## 3.2 Concept-Task Reasoning

- *"Unlike prior CBMs, CREAM does not require all concepts to connect directly to the task."* I am not so sure about that. Doesn't causal CBM [1] do that, too?

## Section 5.2

- Again, isn't CREAM without the side channel identical to [1]? I think it is not fair to claim that CREAM without side channel is computational most efficient if CREAM without side channel is just another existing method.

[1] Dominici, Gabriele, et al. "Causal concept graph models: Beyond causal opacity in deep learning." International Conference on Learning Representations (2025).

---

> ### Author Response · Authors · 2026-06-22
>
> Thank you for the careful review and for the insightful comments and suggestions. We appreciate the time and effort invested in evaluating our work, and we address each point below.
>
> > The proposed architecture is similar to causal CGM. The only difference to CGM is the side channel. Is the efficiency claim without a side-channel fair?
>
> We acknowledge the conceptual similarity with [1], as both methods incorporate structured relationships between concepts. However, CREAM differs in two important respects. First, it is substantially more computationally efficient, making it able to train on datasets where CGM becomes prohibitively resource-intensive. Second, we quantify the influence of the side-channel, enabling assessment of how much model performance is attributed to concepts. These differences stem from fundamentally different modeling paradigms:
> - Concept representation. **CREAM uses CBMs, whereas CGM uses Concept Embedding Models [2]**. That is, while CREAM uses scalar concepts (e.g. binary or probabilistic), and supports mutual exclusivity, CGM uses high-dimensional concept embeddings split into positive and negative components, that do not enforce mutual exclusivity, lead to concept leakage [3] and reduced interpretability.
> - Optimization. CGM relies on graph duplication and multiple specialized losses, whereas CREAM uses cross-entropy and masked layers (StrNN).
> - Computational cost. CREAM has CBM-like GPU efficiency, while CGM depends on CPU-only graph operations making it slower and far more memory-intensive (≥40× more memory than CREAM; Table 10). On CUB, CGM did not complete a single epoch on a 96-core CPU with 256 GB RAM.
>
> > With the side channel, interpretability is somewhat obsolete and the focus on the concepts is hard to quantify. Doesn’t the side channel eliminate the bottleneck?
>
> We believe this concern stems from viewing the side-channel as bypassing the bottleneck. In CREAM, the bottleneck remains; information is partitioned into a supervised pathway (concepts) and an unsupervised pathway (side-channel), similar to prior CBM variants that incorporate auxiliary information [4,5], which are still generally regarded as CBMs.
> The side-channel does not preclude interpretability because its contribution is both controllable and measurable. It is optional (Tables 2 and 4 report CREAM without it), can be suppressed during inference through dropout, and its influence is quantified via CCI. Consequently, CREAM does not treat interpretability as a binary property but as a spectrum: a model with CCI = 0.9 is more concept-grounded than one with CCI = 0.1 and more interpretable than a black-box.
>
> Architecturally, each side-channel neuron is tied to a single class and acts as a class-specific correction. It captures class-discriminative information; e.g., on iFMNIST, the concepts {Clothes, Tops} cannot separate T-shirt, Pullover, and Shirt, so the side-channel encodes the remaining signal. Consistently, Appendix E.5.1 shows near-zero side-channel variance for fully concept-determined classes (e.g., Dress) and substantially higher variance for concept-incomplete classes (e.g. T-shirt).
>
> Finally, as discussed in Future Work, the interpretability of the side-channel can be further improved via concept discovery [6].
>
> > What does "reasonable" mean in the title?
>
> 'Reasonable' is used in a dual sense: (i) interpretable, graph-grounded predictions, and (ii) practical, efficient models that work under incomplete concepts. If the term remains unclear, we are open to revising the title (e.g. “Towards Reasoning Concept Bottleneck Models.”)
>
> > Interventions and the side-channel
>
> The side-channel is structurally independent of the concept pathway (Section 4.3). Thus, by design, interventions affect only the concept-to-task route, while the side-channel remains at its predicted value.
>
> > Mention the C−C and C→Y relationships in Fig. 1.
>
> We updated the figure's caption to explicitly show the C–C and C→Y edges.
>
> > 'Unlike prior CBMs, CREAM does not require all concepts to connect directly to the task'
>
> The statement refers to standard CBMs, where all concepts are fully connected to all tasks. To avoid ambiguity, we rephrase as “prior fully-connected CBMs”.
>
> [2] Espinosa Zarlenga, Mateo, et al. Concept embedding models: Beyond the accuracy-explainability trade-off. Advances in neural information processing systems 35 (2022)
> [3] Parisini, Enrico, et al. Leakage and interpretability in concept-based models. arXiv preprint arXiv:2504.14094 (2025).
> [4] Havasi, M., Parbhoo, S., & Doshi-Velez, F.  Addressing leakage in concept bottleneck models. Advances in Neural Information Processing Systems, 2022.
> [5] Sinha, S., & Zhang, A. . A Comprehensive Survey on the Risks and Limitations of Concept-based Models. arXiv:2506.04237 (2025).
> [6] Fel, Thomas, et al. Craft: Concept recursive activation factorization for explainability. Proceedings of the IEEE/CVF conference on computer vision and pattern recognition. 2023.

---

> > ### Comment · Reviewer_guQj · 2026-06-25
> >
> > I thank the Authors for the clarifications and the revision. From my side, the manuscript can be accepted now.

---

### Review · Reviewer_7YQ9 · 2026-05-07

**Summary Of Contributions:**

Concept Bottleneck Models enable explicit representations of learned models that enable transparent reasoning and human intervention.  In this paper, Concept REAsoning Models (CREAMs) architecturally represent Concept-Concept such as mutual exclusivity / hierarchical / correlation and sparse Concept-Label relationships.

Strengths:
* Architecture - The proposed model seems more amenable to intervention and improved accuracy.
* Intervention Results - The proposed model seems to be amenable to interventions.

Weaknesses:
* To me the clearest weakness of the paper is the lack of clarity / guidance of which method should be used under which settings. Eg., if I care about (1) leakage free most or (2) care about responsiveness to intervention most or (3) care about accuracy most, and I have data of a certain variety, it is quite hard for a reader of the paper to understand the conclusions without joining a lot of information in the results and spending time drawing conclusions. I think this is a key value add for the paper because the method is quite simple and readers would be keenly interested in the empirical tradeoffs it provides. Otherwise, I think it is quite hard for the uninformed reader to make use of the method.
* To elaborate: across the different datasets it seems that the slope of accuracy vs intervention changes across methods changes, while at the same time the ranking of overall accuracy does too. Methods also have certain properties that make this so. The analysis of CREAM itself is nice, but I think there needs to be more comparisons to the other methods.
* I would also encourage the authors to put more examples of the correlation matrices and the learned structures into the paper. I would do it for each dataset. I would use it as an exposition tool when presenting CREAM to the readers.

**Audience:**

Yes

**Audience Explanation:**

Yes, though, I think making the paper more accessible to those less familiar with CBMs would be good.

**Broader Impact Concerns:**

None.

**Claims And Evidence:**

Yes

**Claims Explanation:**

Mostly yes, but I think that the presentation of the trade-offs between the proposed method and the baselines is not clear.

**Requested Changes:**

Clearer guidance on the pros/cons of the proposed method and when to use CREAM compared to its baselines.

---

> ### Author Response · Authors · 2026-06-22
>
> Thank you for the careful review and for the insightful comments and suggestions. We appreciate the time and effort invested in evaluating our work, and we address each point below. All changes introduced in the revision are summarized in an aggregated comment addressed to all reviewers.
> > Clearer guidance on the pros/cons of the proposed method and when to use CREAM compared to its baselines.
>
> We agree that clearer guidance on the strengths, limitations, and intended use cases of the different CREAM options would be valuable. To address this, we have added a new section “Practical considerations” (now Section 6), which discusses when different options are most applicable.
>
> The guiding principle is to select an efficient model that (i) matches the distributional structure of the concepts (independent, correlated, hierarchical, or mutually exclusive), (ii) minimises concept leakage to preserve the reliability of explanations, and (iii) achieves the desired trade-off between performance and interpretability. Among the models considered in this work, CREAM is the only framework that can tackle all of the above.
> Specifically in the new section we discuss:
> - **Availability of structural knowledge.** The more prior knowledge is available, the better it can be exploited. When fully available, it can be encoded directly as a reasoning graph. When only partially available, known relations can be specified explicitly while unknown parts can be left densely connected (e.g., fully bidirectional), allowing the model to learn missing structure. When no prior structure is available, it can be partially discovered automatically using structure learning methods (e.g., CREAM with causal discovery; Appendix E.6). When prior knowledge is available, interpretability is further improved by incorporating it into the model structure.
> - **Concept representation.** Both soft- and hard-concept variants (H-CREAM; Appendix G) are applicable only for discrete concepts and cannot directly handle real-valued concepts; in such cases, logits can be used. For categorical concepts, CREAM additionally supports mutually exclusive groups via group-wise softmax. In general, soft concepts tend to achieve better performance than hard concepts due to easier optimization and their ability to represent uncertainty. Concept leakage can be reduced through several design choices: hard-concept models (H-CREAM) reduce leakage, as do structured graph-based models and softmax-based concept formulations (Table 4) used in CREAM.
> - **Ease of interventions.** Models with explicit graph structure, such as CREAM, facilitate error tracing and enable more targeted interventions. Unlike vanilla CBMs, structured models can propagate interventions through learned or specified relationships. If intervenability is prioritized over predictive performance, one can enforce invertibility in the C-C block (setting $d_C=1$), although this may reduce overall accuracy (Fig. 19). When a side-channel is used, higher CCI values generally lead to stronger intervention effects (Fig. 8, right). Finally, hard-concept models further simplify interventions by reducing them to binary state changes, avoiding heuristic choices such as selecting activation thresholds (e.g. the 5th and 95th percentiles).
> - **Concept set completeness.** When the concept set is incomplete, practitioners must either accept a performance ceiling or use a regularised side-channel to recover missing predictive information (e.g., CREAM, CBM+SC). In general, the more incomplete the concept set, the stronger the side-channel regularization should be. Stronger regularization increases reliance on concepts, leading to higher CCI values, which in turn improve interpretability and intervention effectiveness. We always recommend maintaining $CCI > 0.5$.
>
> We further emphasize that CREAM is a flexible framework that subsumes existing approaches (Table 1) and supports different reasoning graphs, concept representations, and side-channel configurations within a single model. This flexibility allows practitioners to encode knowledge in different forms, compare alternative specifications, and validate modelling assumptions within the same architecture. As a result, practitioners can systematically explore these trade-offs without switching model classes, while maintaining computational efficiency.
>
> > More examples of the correlation matrices and learned structures.
>
> We agree with the reviewer on including more examples of correlation matrices learned by CREAM. We already have assigned a section about them in “Appendix E.5 - Correlation of variables” where we include some examples of the correlation matrices in different datasets and cases of CREAM, as well as covariance matrices of the side-channel and class. Due to space constraints, we chose to include only the iFMNIST case in the main paper, to help the concept leakage explanation.

---

### Review · Reviewer_MofL · 2026-06-10

**Summary Of Contributions:**

This paper proposes CREAMs (Concept REAsoning Models), a novel framework for Concept Bottleneck Models (CBMs) that integrates reasoning graphs.The primary contributions are summarized as follows:


・By adopting a structured neural network to explicitly model the relationships among concepts, as well as between concepts $C$ and the target task $Y$, the framework efficiently represents concept properties such as mutual exclusivity, hierarchical structures, and correlations. Furthermore, the proposed framework generalizes existing approaches, including standard CBMs.

・By isolating elements that cannot be fully explained by concepts alone into a side-channel and applying appropriate Dropout regularization, the framework maintains high predictive performance without compromising the interpretability of concepts, while also suppressing concept leakage.

・The paper introduces the Concept Contribution Index (CCI), a new evaluation metric based on SAGE (Shapley Additive Global Explanations). This index enables the quantitative evaluation of model interpretability derived from concepts even when a side-channel is present.CREAMs is demonstrated to maintain high predictive accuracy while showing robustness against concept interventions, capturing concept-to-concept correlations, and verifying the regularization effect of Dropout on the side-channel.

**Additional Comments:**

・Defining and annotating concept sets beforehand is generally known to incur substantial practical costs and human effort. How scalable or feasible do you expect the proposed workflow to be in real-world application scenarios, where domain experts must define complex concept-to-concept correlations or hierarchies and accurately annotate each sample accordingly? I would  appreciate the authors' perspective on this practical limitation.


・I find the application of Dropout regularization to the side-channel to be an  effective approach. However, in practice, it is  difficult to know the exact degree of incompleteness of a concept set beforehand, making the selection of an appropriate dropout rate non-trivial. Is there a recommended heuristic, protocol, or efficient tuning method to determine this hyperparameter systematically?

**Audience:**

Yes

**Audience Explanation:**

This paper is highly relevant and interesting to researchers in the Explainable AI (XAI) community, particularly those focusing on Concept-based XAI (C-XAI).

Additionally, because the proposed architecture allows the integration of domain experts prior knowledge such as hierarchical structures or correlations among concepts into the model's reasoning process, this work is expected to attract significant interest from practitioners and applied researchers.

**Broader Impact Concerns:**

No Impact Concerns.

**Claims And Evidence:**

Yes

**Claims Explanation:**

The majority of the claims are well-supported, but there are some concerns and points of ambiguity regarding the experimental setup and baselines:

Some aspects of the experimental configuration and the selected baselines lack clarity and justification.

・The exact architectures and model configurations used for the "Black-box" and "$C_{\text{true}} \to Y$" baselines are not clearly specified.

・Regarding the existing method "ECBM," which the authors claim they were unable to replicate, no literature citation is provided. Moreover, the motivation behind choosing ECBM as a primary baseline for comparison in this study is not sufficiently explained.

**Requested Changes:**

Critical Changes

・As raised in the evaluation section, please clearly specify the exact model architectures and configurations used for the "Black-box" and "$C_{\text{true}} \to Y$" baselines.

・Regarding the "ECBM" baseline that was reported as irreproducible, please provide the proper literature citation and explicitly state the motivation for including it as a comparative baseline in your work.





Changes that Would Strengthen the Work

・It might be difficult to intuitively understand what kind of information is practically expected to flow through the side-channel. Similar to how you illustrated concept-to-concept mutual exclusivity (e.g., "Clothes," "Goods") in Figure 1, please provide concrete examples showing the relationship between the side-channel, the concepts, and the target task $Y$.

・In Section 5.2, you claim that even when concepts are significantly removed, CCI does not decrease due to the regularization of the side-channel. However, since the horizontal axis of the left plot in Figure 9 is labeled "Number of concepts," this effect is not immediately intuitive. I recommend changing the horizontal axis to "Concept reduction percentage" (or a similar relative metric) to more clearly and effectively support your claim.

---

> ### Author Response · Authors · 2026-06-22
>
> Thank you for the careful review and for the insightful comments. We appreciate the time and effort invested in evaluating our work, and we address each point below. All changes introduced in the revision are summarized in a comment addressed to all reviewers.
>
> > Specify the model architectures for the "Black-box" and '$C_{true} → Y$' baselines.
>
> We provide descriptions in Section 5.1. The black-box model shares the same frozen backbone as all CBMs, with a single classification head. As mentioned in Appendix D, $C_{true} → Y$ is a linear classifier trained on ground-truth concepts. We also tested a 3-layer MLP on iFMNIST and CelebA, which gave negligible improvements.
>
> > Provide the citation for ECBM and state the motivation for including it as a comparative baseline.
>
> We added the citation [1] and a brief motivation in Section 5.1. ECBM is included as it models concept–concept and concept–class dependencies implicitly via a Boltzmann-based joint energy, over concepts and class labels, instead of an explicit graph. This enables conditional modelling such as $p(c_k | c_{k'})$ and $p(c_k | y, c_{k'})$, and supports interventions such as predicting remaining concepts given a corrected one, making it a natural comparison for CREAM's explicit graph-structured approach. But, we were unable to obtain satisfactory results when using our backbones.
>
> > Relationships between the side-channel, the concepts, and the task.
>
> Architecturally, each side-channel neuron is tied to a single class and acts as a class-specific correction. It captures class-discriminative information; e.g., on iFMNIST, the concepts {Clothes, Tops} cannot separate T-shirt, Pullover, and Shirt, so the side-channel encodes the remaining signal.
>
> To better understand this behavior, App. E.5.1 visualizes the variance of side-channel outputs. Fig. 17 shows near-zero variance in the side-channel neurons corresponding to fully concept-determined classes (e.g., Trouser, Dress) and substantially higher variance in the neurons associated with concept-incomplete classes (e.g., T-shirt, Shirt, Ankle Boot). Moreover, in Fig. 15 we further show correlations between the side-channel exogenous variables and concepts, and observe no clear structure.  As discussed in Future Work, concept discovery [2] could be applied to assign semantic meaning to these "residual" representations.
>
> > Changing Figure 9 to more clearly support the claims.
>
> We updated Fig. 9 to display ‘Removed Concepts (%)' rather than concept count, to better convey the message that CCI does not rapidly decrease with the number of concepts removed.
>
> > Scalability of defining C-C and C->Y relationships.
>
> Scalability is a concern shared by all CBMs. In practice CREAM can help mitigate human effort:
>
> *Data Annotation*. Hierarchical or mutex structures simplify annotation by constraining valid concept assignments during labeling.
>
> *Structure*. The reasoning graph can mirror the structure when annotating. CREAM also supports partial specification, allowing users to encode known relations while leaving others unconstrained (e.g. dense bidirectional connections), with the side-channel compensating for missing concepts. Graphs can be reused across tasks, amortizing their cost. Moreover, CREAM supports different reasoning graphs, thus practitioners can efficiently test and compare alternative structural hypotheses. Finally, App. E.6 provides an example of an automatically discovered graph via causal discovery, reducing reliance on expert-designed structures.
>
> > Selecting the dropout rate for the side-channel
>
> We agree that selecting the dropout rate ($p$) depends on the dataset and concept set and thus, based on our experiments, we suggest the following protocol:
>
> - **Estimate concept completeness**. Train a $C_{true} → Y$ model; its accuracy provides an upper bound given the concept set. A small gap to 100% indicates that the concept set is nearly complete and lower $p$ suffices, whereas a larger gap suggests the need for higher $p$. This is shown in Fig. 8 and Fig. 10.
> - **Use CCI for tuning**. Sweep $p$ and select a value that achieves high task accuracy while maintaining $CCI>0.5$, ensuring a meaningful contribution from the concepts. In our experience, $p≈0.8$ serves as a strong default.
> - **Dynamic adjustment (mentioned in Future Work)**. Perform annealing of $p$ during training based on performance: when task performance remains low despite high concept accuracy, reduce $p$ to increase side-channel influence; as concept predictions stabilize and performance improves, increase $p$.
>
> [1] Xu, Xinyue, et al. "Energy-based concept bottleneck models: Unifying prediction, concept intervention, and probabilistic interpretations.” In The Thirteenth International Conference on Learning Representations (ICLR), 2024.
>
> [2] Fel, Thomas, et al. "Craft: Concept recursive activation factorization for explainability." Proceedings of the IEEE/CVF conference on computer vision and pattern recognition. 2023.

---

> > ### Comment · Reviewer_MofL · 2026-06-29
> >
> > Thank you for the rebuttal and the revisions. I have two additional questions.
> >
> > $\textbf{1. Issues with ECBM}$
> >
> > I understand that ECBM did not achieve satisfactory results with the backbone used in this work, but what is the specific cause? If this is due to the problem setting, is it possible that CREAM would also underperform in settings where ECBM works well?
> >
> > $\textbf{2.Prior domain knowledge and LLMs}$
> >
> > The need for prior domain knowledge appears to be a limitation of this work. Could leveraging LLMs to infer the $C-C$ and $C→Y$ relationships potentially address this issue? If so, it might be good to mention this in the future work.

---

> > > ### Author Response · Authors · 2026-07-03
> > >
> > > We thank the reviewer for the additional comments and for the helpful feedback. We have incorporated the suggestions into the revised manuscript. **All changes are highlighted in purple.**
> > >
> > > > Issues with ECBM
> > >
> > > We invested significant effort in integrating ECBM into our unified experimental pipeline but were unable to reproduce competitive task performance on CUB and CelebA, although ECBM achieved satisfactory performance on both FMNIST variants. We believe this is primarily due to differences between our standardized evaluation protocol, used consistently across all baselines, and the original ECBM experimental setup.
> > >
> > > **Experimental setup differences.**
> > > To ensure a fair comparison across all baselines, all methods were evaluated under the same protocol, including identical backbones, preprocessing, image resolutions, and batch sizes. For example,  we set the batch size to 256 across all datasets, to match our experimental section, while the original ECBM work uses a batch size of 64; and for CelebA we used our subset of concepts instead of all available concepts. Moreover, instead of the ResNet101 backbone all baselines use: ResNet-18 for CUB and CelebA, and a small CNN for iFMNIST and cFMNIST. We note that the energy-based inference in ECBM relies on iterative gradient-based optimization at test time to jointly recover concept probabilities and class predictions by minimizing the joint energy; this procedure may be particularly sensitive to the backbone, resolution, and batch size, which could explain part of the gap we observed.
> > >
> > > **Regarding whether CREAM might underperform in settings where ECBM excels.**
> > >
> > > ECBM's main strength lies in its flexible probabilistic framework, which does not require a pre-specified reasoning graph. This makes it well suited to settings where concept–concept and concept–class relationships are unknown or difficult to elicit. Moreover, ECBM allows for propagation of interventions. In contrast, CREAM, while it does not easily propagate interventions since invertibility is required, is designed to explicitly exploit structural knowledge, enabling domain expertise to be incorporated directly into both reasoning and interventions. Consequently, CREAM is particularly well suited not only to domains where such knowledge is available, but also to scenarios in which practitioners wish to specify, compare, and validate alternative reasoning structures corresponding to different domain hypotheses. Another distinction concerns interventions: ECBM's joint energy formulation naturally propagates interventions across correlated concepts without requiring invertibility, whereas CREAM currently emphasizes explicit, transparent reasoning through a user-specified causal structure. We therefore view the two approaches as addressing complementary application scenarios with different design trade-offs, rather than one strictly dominating the other.
> > >
> > >
> > >
> > > > Prior Domain Knowledge and LLMs
> > >
> > > We agree that this is a promising direction. Recent work in the structure learning community has begun leveraging LLMs to assist causal discovery algorithms, improving both the quality and scalability of inferred structures [1,2]. Separately from causal discovery, LLMs could also be used as standalone knowledge elicitation tools to directly infer both the concept-concept ($C-C$) and concept-task ($C \rightarrow Y$) relationships from domain knowledge or textual descriptions, reducing the need for manual specification.
> > >
> > > Following the reviewer's suggestion, we now highlight the potential of both LLM-assisted causal discovery and direct LLM-based graph specification as future research directions in the Future Work and Limitations section, emphasizing their potential to improve the practical scalability of CREAM.
> > >
> > > [1] Emre Kiciman, Robert Ness, Amit Sharma, and Chenhao Tan. Causal reasoning and large language models:
> > > Opening a new frontier for causality. Transactions on Machine Learning Research, 2024. ISSN 2835-8856.
> > > URL https://openreview.net/forum?id=mqoxLkX210. Featured Certification
> > >
> > >
> > > [2] Jing Ma. Causal inference with large language model: A survey. Findings of the Association for Computational
> > > Linguistics: NAACL 2025, pp. 5886–5898, 2025.

---

> > > > ### Comment · Reviewer_MofL · 2026-07-07
> > > >
> > > > Dear Authors,
> > > >
> > > > Thank you for your clear explanations.
> > > > I understand the differences between ECBM and CREAM, and how to apply them in practice.
> > > > I also appreciate the added text about LLMs and prior domain knowledge.
> > > >
> > > > All my concerns are addressed

---

### Author Response · Authors · 2026-06-22
**Summary of all changes to manuscript**

We thank all reviewers for their careful reading, constructive feedback, and the time invested in evaluating our work. We have addressed each comment in detail in our individual responses, and summarize below the corresponding changes made to the manuscript. ***Changes in the manuscript are highlighted in blue.***
# Key Improvements
## Clarity of methodology and baselines - Section 5.1; Model Baselines
We clarified the architectures of the black-box and $C_{true}→ Y$ baselines to improve reproducibility and readability. We also added the ECBM baseline citation and mentioned clarified its inclusion criteria. - ***Reviewer MofL***
## Clarifying side-channel’s interpretation - Section 5.2; Side-channel and (in)complete settings
We further clarified the role of the side-channel, which consists of class-specific neurons that capture residual class-discriminative information not explained by the concept set. For concept-complete classes, its activations are nearly constant, while for concept-incomplete classes it becomes more variable, indicating that it compensates for missing concept information (e.g., distinguishing T-shirt, Pullover, and Shirt in iFMNIST). This behavior is supported empirically in App. E.5.1, where we observe low variance for fully concept-determined classes and higher variance otherwise, with no clear alignment between side-channel variables and existing concepts.
Importantly, the side-channel does not bypass the bottleneck but complements it, splitting information into a supervised concept pathway and an unsupervised residual pathway. Its contribution is controllable via dropout and quantified by CCI, enabling a spectrum of interpretability rather than a binary notion. As noted in Future Work, its semantics could be further enhanced using concept discovery methods. - ***Reviewers MofL, guQj***
## New section: Practical considerations
We added a new “Practical Considerations” section (now Section 6)  highlighting one of CREAM’s main advantages: a unified and efficient framework that supports different reasoning graphs, concept representations, and side-channel configurations. This allows practitioners to systematically explore design trade-offs and evaluate alternative hypotheses without changing architectures.  - ***Reviewer 7YQ9***

In short, the section provides guidance on selecting among configurations:

**Structural knowledge**. We discuss how different levels of prior knowledge can be incorporated into the reasoning graph. For example, when knowledge is only partially available, known relations can be specified explicitly while unknown relations remain densely bidirectionally connected, allowing the model to learn the missing structure. When no prior structure is available, it can be partially discovered automatically using structure-learning methods (e.g., causal discovery; Appendix E.6).

**Concept representation and leakage**: We discuss when different concept representations are applicable, the trade-offs between soft and hard concepts, the handling of mutually exclusive concepts, and practical strategies for reducing concept leakage.

**Ease of interventions**: We discuss the trade-off between intervention propagation and predictive performance, including the option of enforcing an invertible concept-concept block to support intervention propagation through the reasoning graph. We also discuss how side-channel regularization influences intervention effectiveness.

**Concept completeness and the side-channel**: We discuss when a side-channel is beneficial, how concept completeness can be estimated, and provide practical guidance for selecting the side-channel dropout rate based on the degree of concept incompleteness, including a recommended default value. - ***Reviewer MofL***

Overall, this section provides concrete guidance on matching model design choices to practitioner priorities in interpretability, predictive performance, intervention behavior, and available domain knowledge.
# Minor improvements:
- Section 1: Updated Fig. 1 caption to explicitly label both $C$–$C$ and $C → Y$ relationships. - ***Reviewer guQj***
- Section 3: (i) Clarified the meaning of the term reasonable and its intended interpretation. (ii) Replaced “prior CBMs” with “prior, fully-connected CBMs” to avoid ambiguity in Section 3.2. - ***Reviewer guQj***
- Section 4.3: (i) Clarified that the side-channel does not affect interventions  - ***Reviewer guQj***,
(ii) Added a note on selecting the dropout rate, with pointers to the experimental and guideline sections  - ***Reviewer MofL***
- Section 5: Updated Fig. 9 to show ‘Removed Concepts (%)' instead of concept count, for improved readability. - ***Reviewer MofL***
- (now) Section 7 - Future Work and Limitations: Added comments on the scalability of CREAM in practice. - ***Reviewer MofL***